# A STATISTICAL FRAMEWORK FOR EFFICIENT OUT OF DISTRIBUTION DETECTION IN DEEP NEURAL NETWORKS

**Matan Haroush**[*]
Electrical & Computer Engineering Department
Technion, Haifa, Israel.
Habana Labs - An Intel company
mharoush@habana.ai

**Tzviel Frostig**[*]
Statistics & Operation Research Department
Tel Aviv University, Israel.
tfrostig@gmail.com

**Ruth Heller**
Statistics & Operation Research Department
Tel Aviv University, Israel.
ruheller@tauex.tau.ac.il

**Daniel Soudry**
Electrical & Computer Engineering Department
Technion, Haifa, Israel.
daniel.soudry@gmail.com

## ABSTRACT

**Background.** Commonly, Deep Neural Networks (DNNs) generalize well on samples drawn from a distribution similar to that of the training set. However, DNNs' predictions are brittle and unreliable when the test samples are drawn from a dissimilar distribution. This is a major concern for deployment in real-world applications, where such behavior may come at a considerable cost, such as industrial production lines, autonomous vehicles, or healthcare applications.

**Contributions.** We frame Out Of Distribution (OOD) detection in DNNs as a statistical hypothesis testing problem. Tests generated within our proposed framework combine evidence from the entire network. Unlike previous OOD detection heuristics, this framework returns a $p$-value for each test sample. It is guaranteed to maintain the Type I Error (T1E - incorrectly predicting OOD for an actual in-distribution sample) for test data. Moreover, this allows to combine several detectors while maintaining the T1E. Building on this framework, we suggest a novel OOD procedure based on low-order statistics. Our method achieves comparable or better results than state-of-the-art methods on well-accepted OOD benchmarks, without retraining the network parameters or assuming prior knowledge on the test distribution — and at a fraction of the computational cost.

## 1 INTRODUCTION

Deep Neural Networks' (DNNs) predictions were shown to be overconfident and unreliable when the test samples are drawn from an unexpected distribution (e.g., Biggio et al. (2013); Szegedy et al. (2013); Goodfellow et al. (2014); Eykholt et al. (2017)). Therefore, numerous Out Of Distribution (OOD) detection methods were suggested to address this issue by informing an operator when not to trust the DNN's prediction.

In the context of DNNs, methods for OOD detection can be roughly divided into two classes. **"Mutators"** methods modify the network structure or loss, and depend on training its parameters to provide a confidence measure (Hendrycks et al., 2018; Malinin and Gales, 2018; Winkens et al., 2020; Zhang et al., 2020; Hsu et al., 2020). **"Observer"** methods model the prediction uncertainty of a pre-trained network, without modifying its architecture or parameters (Hendrycks and Gimpel, 2016; Liang et al., 2017; Lee et al., 2018; Zisselman and Tamar, 2020; Raghuram et al., 2020).

Mutators are potentially costly, as they require substituting or retraining the network and subsequently repeating an exhaustive hyperparameter tuning process. Also, the resulting model quality on the

---

[*]Equal contribution

original task may vary. Alternatively, observer methods commonly utilize auxiliary models to monitor features produced by the reference network. Since this approach is decoupled from the network optimization process, the reference accuracy is retained at the cost of additional test time resources, such as compute and memory.

Another differentiating factor between methods, regardless of their class, stems from fundamental assumptions regarding prior knowledge on the test distribution. Specifically, some methods (e.g., Hendrycks et al. (2018); Lee et al. (2018); Zisselman and Tamar (2020)) rely on prior knowledge, such as access to the OOD data or its proxy. However, this prior can lead to undesirable bias and poor detection performance on other test distributions.

Notably, OOD detection can be naturally cast as a statistical hypothesis testing problem (Grubbs, 1969; Vovk et al., 2005; Papadopoulos, 2008; Grosse et al., 2017). Nevertheless, authors of contemporary observer methods largely overlook the statistical aspects of their proposed algorithms. For example, they do not provide meaningful error guarantees nor explicitly define what hypothesis is being tested. For further details regarding related work, see Section 5. In addition, these methods also tend to be computationally expensive (see Section 4.4). In this work, we aim to tackle these issues.

**Contributions.** We first propose methodological innovations

- We present a novel framework for OOD detection in DNNs, based on statistical hypothesis testing and phrase two relevant null hypotheses. The first is whether a sample is drawn from the same distribution of the classes in the training dataset (any-class). The second is whether a sample is drawn from the distribution corresponding to the predicted class (class-conditional). We construct valid tests for both hypotheses and further provide empirical evidence for these theoretical results (Section 3).

- Our framework can be viewed as an extension of the inductive conformal predictor framework (Papadopoulos, 2008), which is a non-parametric method for conducting hypothesis testing with an arbitrary test statistic (Vovk et al., 2005). However, instead of focusing solely on the last layer of the DNN predictor, we show how to normalize and aggregate multiple non-conformity (i.e., abnormality) measures based on the entire network. Empirically, we demonstrate that this generalization is crucial for OOD detection. Namely, we find that a multi-layer detector improves accuracy by 50% on average compared to single-layer variants (Section 4.2).

These innovations lead to considerable practical advantages

- Importantly, unlike current methods for OOD detection in DNNs, our framework returns a valid $p$-value for each tested sample. This guarantees maintaining a Type I Error (detecting OOD as in-distribution) on test data (see Section 3.2, Propositions 1 and 2), and has additional advantages[1] which do not exist in standard threshold-based OOD classification.

- Based on this framework, we design a detection scheme denoted as MaSF (Max-Simes-Fisher, Section 4.2). MaSF achieves similar or better results compared to SOTA on the benchmark proposed by Lee et al. (2018), while in our experiments, we show its test statistic inference is 35 times faster. Making MaSF appealing for real-world applications with a limited compute budget (Section 4.4).

- We demonstrate potential avenues to further reduce the computational overhead via structured sparsity (i.e., channel sampling) — further reducing the computational cost by a factor of 10 while degrading accuracy by no more than $\sim 1\%$ (Section 4.5).

## 2 BACKGROUND

In the following section, we provide a summary of Null Hypothesis Statistical Testing (NHST). We begin with a single hypothesis and then further expand on Global-Null tests.

**Null Hypothesis Statistical Testing.** NHST is the process of choosing between two hypotheses - $\mathcal{H}_0$ (null) and $\mathcal{H}_1$ (alternative) regarding observation (sample) $X$. It is formalized as $\mathcal{H}_0 : X \sim \mathcal{P}$ vs

---

[1] For example, $p$-values enable adjusting the decision rule when more than one hypothesis is tested at a time while maintaining a certain error criterion (Chapter 2 in Bretz et al. (2016)). For instance, one can potentially employ several OOD detectors on a single sample (e.g., each specialized on a specific type of outlier distribution).

$\mathcal{H}_1 : X \not\sim \mathcal{P}$, where $\mathcal{P}$ is some distribution. The decision to reject $\mathcal{H}_0$ is based on the critical value $\gamma$ and the test statistic $T(X)$. Throughout this section, we will focus on right-tailed tests. We call a test right-tailed when higher values of $T(X)$ are more likely under $\mathcal{H}_1$ than under $\mathcal{H}_0$. In those kinds of tests, $\mathcal{H}_0$ is rejected if $T(X) \geq \gamma$. The critical value $\gamma$ quantifies the required evidence to determine if $\mathcal{H}_0$ is false.

Erroneously rejecting $\mathcal{H}_0$, is called a Type I Error (T1E). The significance level (i.e., probability of T1E) is given by $\alpha = P_{\mathcal{H}_0}(T(X) \geq \gamma)$. Usually, $\alpha$ is predetermined, and $\gamma$ is found accordingly. For a given significance level, the goal is to maximize the power of the test, $P_{\mathcal{H}_1}(T(X) \geq \gamma)$, which is the probability of correctly rejecting $\mathcal{H}_0$. Commonly, a $p$-value is used to summarize NHST. The $p$-value indicates the probability of obtaining a test statistic equal or more extreme than the observed test statistic, given that $\mathcal{H}_0$ is true. The $p$-value (denoted by $q$) is given by, $q(t_{\mathrm{obs}}) = P_{\mathcal{H}_0}(T(X) \geq t_{\mathrm{obs}})$, where $t_{\mathrm{obs}}$ is the observed value of the test statistic.

Assuming the null is correct, and that $\mathcal{P}$ is continuous, then the $p$-values are uniformly distributed (Abramovich and Ritov, 2013). This allows to simply reject the null if $q(t_{\mathrm{obs}}) \leq \alpha$. It is clear that the significance level will be maintained if the $p$-value distribution is stochastically larger[2] than or equal to the uniform distribution[3].

**Global Null Tests.** Given a set of hypotheses, combination tests are used to determine if any hypothesis is false (global null). Let $\mathbf{q}$ denote a vector of $m$ $p$-values, $\boldsymbol{q} = (q_1, \ldots, q_m)$, that is used to test the $m$ null-hypotheses, $H_{0,1}, \ldots, H_{0,m}$. The global null hypothesis is defined as $H_{0,\cdot} = \bigcap_{i=1}^{m} H_{0,i}$. The test statistic is $T(\mathbf{q})$. The test power depends on the unknown distribution of the test statistic under $\mathcal{H}_1$.

Two ubiquitous tests of the global-null are the Fisher and Simes tests (Fisher, 1992; Simes, 1986). The Fisher test statistic is $-2 \sum_{i=1}^{m} \log(q_i)$. It is suitable for detecting dense signals due to the summation of p-values. The Simes test statistic is $\min_{i \in \{1, \ldots, m\}} q_{(i)} \frac{m}{i}$, where $(i)$ is the index of the p-values after sorting. It is adept for detection of a sparse signal, as it focus on a single p-value. These complementing attributes are appealing for OOD detection, as we will demonstrate in the following sections. For a more detailed introduction of these tests see Section B.3.

An important benefit of using $p$-values is that they allow maintaining a meaningful error rate. There are many methods employed to maintain such error rates, which are applied on $p$-values (Holm, 1979; Hochberg, 1988; Benjamini and Hochberg, 1995).

## 3 HYPOTHESIS TESTING IN NEURAL NETWORKS

For the sake of simplicity, we focus the next section solely on convolutional neural networks (CNNs) for image classification tasks. The method presented, however, is applicable to any DNN.

### 3.1 PRELIMINARIES

Define $[k] = \{i \in \mathbb{N} : i \leq k\}$ and let $\chi^c = \{(X_i, y_i) : i \in [n_c], y_i = c\}$, where $|\chi^c| = n_c$, $X_i$ is an image and $y_i, \hat{y}_i$ are its true and predicted class (out of $k$ classes). We further split the observations of each class to the CNN training set $\chi_{\mathrm{train}}^c$, and the validation set $\chi_{\mathrm{val}}^c$. The training set is denoted by $\chi_{\mathrm{train}} = \bigcup_{c=1}^{k} \chi_{\mathrm{train}}^c$, and its cardinality is $N_{\mathrm{train}} = \sum_{c=1}^{k} n_c$. We similarly define $\chi_{\mathrm{val}}$ and $N_{\mathrm{val}}$. $\mathcal{P}^c$ is the class distribution of $X_i$ where $y_i = c, c \in [k]$. The distribution of any test statistic under $\mathcal{H}_0$ is required to obtain $p$-values. For that reason we apply the empirical cumulative distribution function (eCDF) to estimate the null distribution of class $c$, function $T$ at $x$ using an observations set $\chi^c \in \{\chi_{train}^c, \chi_{val}^c\}$,

$$\hat{\mathbb{P}}(x; T, \chi^c) = \frac{\sum_{(X_i, y_i) \in \chi^c} I\left(T(X_i) \leq x\right) + 1}{n_c + 1}, \tag{1}$$

where $I$ is the indicator function. The addition of 1 is necessary to ensure that the $p$-values are not equal to 0, as it implies they are rejected at all significance levels. The problem of determining if an

---

[2] $X$ is stochastically larger than $Y$ if $\forall x\ P(X > x) > P(Y > x)$.

[3] When it is unclear if $T(X)$ is expected to be larger or smaller under $\mathcal{H}_1$, one can use two-sided tests, so the $p$-value is $2 \cdot \min[P_{\mathcal{H}_0}(T(X) \geq t_{\mathrm{obs}}), P_{\mathcal{H}_0}(T(X) \leq t_{\mathrm{obs}})]$, which can be viewed as conducting a right and left sided tests and combining the results with Bonferroni correction.

image $X_{\text{test}}$ is OOD, can be formalized as

$$\mathcal{H}_0^* : \exists c : X_{\text{test}} \sim \mathcal{P}^c, \quad \mathcal{H}_1^* : X_{\text{test}} \not\sim \mathcal{P}^c \; \forall c \in [k] \tag{2}$$

By rejecting $\mathcal{H}_0^*$, we conclude that the image is not drawn from any of the class distributions. One can also test if an image is sampled from a specific class distribution,

$$\mathcal{H}_0^c : X_{\text{test}} \sim \mathcal{P}^c, \quad \mathcal{H}_1^c : X_{\text{test}} \not\sim \mathcal{P}^c. \tag{3}$$

Since the true label is unknown at test time, we are interested in the case where the class is assigned according to the CNN prediction, $\mathcal{H}_0^{\hat{y}_{\text{test}}}$. Therefore, rejecting it indicates the image is either misclassified or OOD.

## 3.2 THE TESTING PROCEDURE

The typical CNN ($F$) is composed of $L$ layers. Each layer, $l$, contains $a_l$ channels. The feature map of the $j$'th channel in $l$'th layer is denoted by $F_{j,l} : X \to \mathbb{R}^{h_l \times w_l}$, where $X$ is the input image, $h_l$ and $w_l$ refer to the spatial dimensions of the feature maps at the $l$'th layer. We begin by testing at the channel level. Each channel is summarized to a scalar value using a spatial reduction function, $T^S$,

$$t_{j,l}(X) = T^S(F_{j,l}(X)) \quad , \quad T^S : \mathbb{R}^{h_l \times w_l} \to \mathbb{R}. \tag{4}$$

Therefore, the empirical null distribution of the $t_{j,l}$ at channel $j$ in layer $l$ is $\hat{\mathbb{P}}(x; t_{j,l}, \chi_{\text{train}}^c)$. The $p$-value is denoted by $q_{j,l}^c$ for channel $j$, and by $q_{\cdot,l}^c$ for the entire layer $l$ (of class $c$). We use two-sided $p$-values for the channel reduction, as outliers can appear in both tails of the distribution, see Section D.1. Next we aggregate the evidence from $q_{\cdot,l}^c$ for each layer, by applying a channel reduction function, $T_l^{\text{ch}}$:

$$t_l^c(X) = T_l^{\text{ch}}(q_{\cdot,l}^c), \qquad T_l^{\text{ch}} : [0,1]^{a_l} \to \mathbb{R}. \tag{5}$$

By employing $\hat{\mathbb{P}}(x; t_l^c, \chi_{\text{train}}^c)$, we can recover the layer's $p$-values for class c: $q_1^c, \ldots, q_L^c$. Given $q_1^c, \ldots, q_L^c$ for all layers, the class conditional test statistic of an image is obtained by applying the layer reduction function, $T^L$,

$$t^c(X) = T^L(q_1^c, \ldots, q_L^c), \qquad T^L : [0,1]^L \to \mathbb{R}. \tag{6}$$

The class conditional $p$-value for sample $X$, is obtained using $\hat{\mathbb{P}}(X; t^c, \chi_{\text{val}}^c)$ and denoted by $q^c(X)$, can be used to test $\mathcal{H}_0^c$ (Eq. 3). Note, that $t^c(X_{\text{test}})$ is not exchangeable with $\{t^c(X) : X \in X_{\text{train}}\}$ since the are dependent through the DNN. Using the validation set ensures the exchangeablity of the test-statistics and through it the validity of the proposes test. Algorithm-1 describes the procedure for obtaining class-conditional $p$-value.

We now turn to present the two main propositions regarding the validity of the procedure described above. The procedure is considered valid if it maintains the T1E at the specified significance level[4]. Proof for both propositions and assumptions can be found in the Appendix B.

**Proposition 1** *A level $\alpha$ test of $\mathcal{H}_0^*$ vs. $\mathcal{H}_1^*$ is: reject $\mathcal{H}_0^*$ if $q^{\max}(X_{\text{test}}) = \max\{q^1(X_{\text{test}}), \ldots, q^k(X_{\text{test}})\} \leq \alpha$.*

Since the true class of the image is unknown, the maximum $p$-value ensures the method rejects $\mathcal{H}_0^*$ for a sample only if the evidence is against all classes, i.e., against $\mathcal{H}_0^1, \ldots, \mathcal{H}_0^k$.

When the class is assigned according to the CNN prediction, $P_{\mathcal{H}_0^{\hat{y}_{\text{test}}}}(q^{\hat{y}_{\text{test}}}(X_{\text{test}}) \leq \alpha) \geq \alpha$, since the decision on which hypothesis to test was made based on the data. We can bound this T1E as stated in the next proposition and adjust the $p$-values accordingly to obtain a valid test.

**Proposition 2** *For testing $\mathcal{H}_0^{\hat{y}_{\text{test}}}$ vs. $\mathcal{H}_1^{\hat{y}_{\text{test}}}$ at significance level $\alpha$, the T1E probability is $P_{\mathcal{H}_0^{\hat{y}_{\text{test}}}}(q^{\hat{y}_{\text{test}}} \leq \alpha) \leq \alpha P(\hat{y}_{\text{test}} = y_{\text{test}}) + P(\hat{y}_{\text{test}} \neq y_{\text{test}}).*

---

[4]Note that $\chi_{\text{train}}$ can be used even though it is correlated with the CNN weights since it allows us to leverage more observation. The procedure remains valid since $\chi_{\text{val}}$ is used to calibrate the final test statistic.

Both propositions rely on the assumption that the validation set used to estimate the CDF of the final combination test is independent of the DNN training set, in a similar vein to inductive conformal prediction (Vovk et al., 2005; Papadopoulos, 2008).

### 3.3 SUMMARY AND CHALLENGES

To summarize, we construct our test function in 3 steps. i) Spatial reduction followed by $p$-value extraction per channel. ii) Channel reduction, summarizing resulting $p$-values for each layer. iii) Layer reduction, aggregating $p$-values from all layers into a final $p$-value. For steps (i) and (ii) we approximate the CDF using $\chi_{\text{train}}$, while $\chi_{\text{val}}$ is used for the final step. Transforming each channel's spatial features into a $p$-value ensures that the evidence from all channels is assessed comparably even though their distribution may vary. At each layer, the aggregation of the per-channel results can cause a disparity (layers with more channels will dominate the test statistic value). Transforming the layer test statistics into $p$-values resolves the issue. Ultimately, this hierarchical approach guarantees a fair comparison at each step when pooling evidence from the entire CNN to reject either $\mathcal{H}_0^c$ or $\mathcal{H}_0^*$.

However, combining multiple tests into a single one has its challenges. On the one hand, we want to include as many hypotheses as the rejection of the global null can be caused by any of them.

---

**Algorithm 1:** Class-conditional $p$-values

**Input** : $F, X_{\text{test}}, c$ ; // The network, input image and class of interest

**Input** : $T^S, T^{\text{ch}}, T^L$ ; // Spatial, channel and layer reductions

**Input** : $\hat{\mathbb{P}}(.; t_{j,l}^c, \chi_{\text{train}}^c)$ $j \in [a_l], l \in [L]$ ; // eCDF per-channel after $T^S$

**Input** : $\hat{\mathbb{P}}(.; t_l^c, \chi_{\text{train}}^c)$ $l \in [L]$ ; // eCDF per-layer after $T^{\text{ch}}$

**Input** : $\hat{\mathbb{P}}(.; t^c, \chi_{\text{val}}^c)$ ; // eCDF after $T^L$

**Output**: $q^c(X_{\text{test}})$; // Class-conditional $p$-value for $X_{\text{test}}$ and class $c$

**for** $l \in [L]$ **do**
    **for** $j \in [a_l]$ **do**
        $t_{j,l}(X_{\text{test}}) = T^S(F_{j,l}(X_{\text{test}}))$ ;
        // Spatial reduction
        $q_{j,l}^c = \min(\hat{\mathbb{P}}(t_{j,l}(X_{\text{test}}); t_{j,l}^c, \chi_{\text{train}}^c), 1 - \hat{\mathbb{P}}(t_{j,l}(X_{\text{test}}); t_{j,l}^c, \chi_{\text{train}}^c))$ ;
        // $p$-value per-channel, assuming two sided test
    $t_l^c(X_{\text{test}}) = T^{\text{ch}}(q_{1,l}^c(X_{\text{test}}), \ldots, q_{a_l,l}^c(X_{\text{test}}))$ ;
    // Channel reduction
    $q_l^c(X_{test}) = \min(\hat{\mathbb{P}}(t_l^c(X_{\text{test}}); t_{j,l}^c, \chi_{\text{train}}^c), 1 - \hat{\mathbb{P}}(t_l^c(X_{\text{test}}); t_l^c, \chi_{\text{train}}^c))$ ; // $p$-value per-layer

$t^c(X_{\text{test}}) = T^L(q_1^c(X_{\text{test}}), \ldots, q_L^c(X_{\text{test}}))$ ; // Layer reduction

***Return*** $1 - \hat{\mathbb{P}}(t^c(X_{\text{test}}); t^c, \chi_{\text{val}}^c)$); // Assuming a right sided $p$-value

---

On the other hand, as more true null hypotheses are included, the power may decrease. For example, consider the case where the combined $p$-values are independent under the null. In the Fisher test, the more test statistics from true null hypotheses are combined, the greater the noise to signal ratio. It implies that the rejection criteria will be harder to reach. In the Simes test, the sorted $p$-values are multiplied by the number of hypotheses, $m$. Again, implying that adding test statistics from true null hypotheses will lower the power of the test.

## 4 DESIGNING TEST STATISTICS FOR OOD DETECTION

So far, we have described our novel theoretical framework for NHST in DNNs. Before we describe our proposed OOD detection algorithm, we briefly review a popular benchmark used to evaluate it and motivate the rationale behind the specific design choices.

### 4.1 OOD DETECTION BENCHMARK

An accepted benchmark for OOD detection was introduced by Lee et al. (2018). We adhere to the same protocol to evaluate our suggested method. Therefore, we use the same pre-trained image classification CNNs (Huang et al., 2016; He et al., 2016) and datasets. In this benchmark, test samples are drawn from the alternative datasets and presented to the reference CNN. Competing methods output an abnormality score per sample by observing the CNN's activations. Unlike classical NHST, the rejection threshold for the predicted OOD score is determined based on the in-distribution test set, where a threshold is found to ensure a False Positive Rate (FPR). That is, results are reported as the True Positive Rate (TPR - i.e., correct OOD prediction rate) while maintaining an FPR of $5\%$ for

in-distribution validation samples (TPR95). Adjustment to the p-value or the threshold is equivalent. In the following section, to ensure a fair comparison we apply the same method in for MaSF.

For comparability of results, we modify our procedure to estimate the CDFs of the null distributions using only the CNN training set. This violates the statistical guarantees of maintaining T1E due to the dependency between the CNN weights and the training set samples. However, since demonstrating the procedure's ability to maintain T1E is of independent interest, we provide additional experiments employing a hold-out set in the Appendix B.2. There, we show OOD detection based on a valid procedure does not change meaningfully from the reported results. Finally, to evaluate the overall performance, we calculate the mean TPR95 (mTPR) and standard deviation (SD), along with the minimal TPR95 (Min-TPR) observed over all test scenarios.

## 4.2 DESIGN CONSIDERATIONS

We now turn to discuss our OOD algorithm, which returns a class conditional $p$-value (i.e., Hypothesis $\mathcal{H}_0^{\hat{y}}$ from Eq. 3) provided the trained network, input image, and the predicted class.

**Choosing reductions.** The choice of reduction functions can greatly impact the power of the final test statistic and incorporate domain knowledge into the designed detector. However, it is not a trivial choice. Therefore, we construct a set of detectors employing simple building blocks and evaluate their overall performance. Specifically, we use Simes and Fisher tests as the channel or the layer reductions along with Max and Mean pooling as spatial reductions. Table 1 portrays the mean and SD of each configuration (see Section 4.1).

Table 1: Hierarchical reduction ablation.

| Layer | Channel | Spatial | mTPR$^{\uparrow}$ | SD |
|---|---|---|---|---|
| Fisher | Simes | Max | 96.4 | 4.7 |
| Fisher | Fisher | Max | 95.3 | 4.9 |
| Simes | Simes | Max | 95.3 | 5.9 |
| Fisher | Simes | Mean | 91.1 | 8.6 |
| Simes | Fisher | Max | 90.1 | 10.7 |
| Fisher | Fisher | Mean | 89.8 | 8.8 |
| Simes | Simes | Mean | 87.9 | 9.5 |
| Simes | Fisher | Mean | 80.1 | 14.3 |

$^{\uparrow}$Results are sorted according to the mean TPR95 (mTPR) on OOD benchmark (Section 4.1)

We observe several consistent trends across all scenarios in the benchmark. Namely, the maximum value of a given feature map is more sensitive to outliers compared to the mean value, which is expected. Additionally, we find that the Fisher combination test performs better than Simes as a layer reduction. This suggests the evidence of abnormality tends to propagate throughout the networks' layers. Finally, the Simes test appears to be a better channel reduction. Note that using Simes (or any other multiple comparison methods) alleviates the need of estimating the distribution at the channel level, as it normalizes the p-value to the number of channels (see Appendix E). Furthermore, it hints that the OOD evidence is either sparse (i.e., abnormality exists in a few of the channels) or that channels are strongly correlated, rendering the Fisher test uninformative.

Moving forward, we focus on the best configuration: Maximum (spatial), Simes (channel), and Fisher (layer) configuration, dubbed MaSF. The MaSF algorithm is available in Appendix F.

**Observing multiple layers.** Next, we aim to investigate the impact of utilizing multiple layers versus relying on a single layer from the end of the model (as in the inductive predictor framework of Papadopoulos (2008)). We evaluate mTPR when using the baseline Maximum Softmax Probability (MSP - Hendrycks and Gimpel (2016)), and the MaSF detector using features from a single layer. Specifically, we consider either the inputs or outputs of the linear classifier or the input feature of the penultimate layer (i.e., the final Average Pooling layer - AP). This is done by eliminating the Fisher layer reduction step. The detector achieved the best results in terms of mTPR (SD) using the AP layer's inputs - 64.7 (21.5). However, these are significantly lower compared to standard MaSF when tracking all layers 96.4 (4.7) as reported in Table 1.

Despite the potential gains from observing additional shallow layers, the test power could decrease when adding non-informative layers (see Section 3.3), and the computational cost increases. Choosing which layers to include in the test is not trivial as it may vary depending on the network architecture, downstream task, or computational budget. Thus, in all our experiments, we test the outputs of all convolution and dense layers in the network. In Appendix G, we experiment with assigning a higher weight to the final layer of the networks. This approach yields improved near distribution results, we leave its development for future work. This is particularly interesting if we know that the OOD

samples are drawn from a distribution close to that of the training data. In this case, we expect shallow layers (i.e., close to the input) to produce features that will not be discriminative.

Finally, in Appendix D, we provide additional analysis on the correlation between the test statistics among the monitored layers. We suggest that variance reduction techniques should play a role in designing new algorithms within our framework. For example, by introducing random sampling or whitening strategies into the construction of the test statistics, we defer this topic for future work.

## 4.3 EMPIRICAL EVALUATION

In this section, we provide a breakdown of MaSF performance on the popular benchmark described in Section 4.1. In addition, Section 5 contains a review of the related work discussed below.

We compare our results with Deep Mahalanobis (Lee et al., 2018), ResFlow (Zisselman and Tamar, 2020) and GRAM (Sastry and Oore, 2019). To the best of our knowledge, these are the best performing observer methods (i.e., they do not retrain the model) to date. We also report MSP as a baseline (Hendrycks and Gimpel, 2016). We omit results tuned using OOD data to provide a fair evaluation. Such knowledge can be integrated within our framework, given the test distribution, by selecting which layers and channels to monitor based on their discriminative power. Additionally, Area Under Receiver Operating Characteristics (AUROC scores and ROC) are provided in the Appendix (Section. G.3 & Fig. 8).

The results for each method are summarized in Table 2. GRAM and MaSF outperform the competing methods in almost all scenarios. Specifically, MaSF has an advantage considering the overall quality (i.e., mTPR, SD, and Min-TPR95), and it is much lighter, as we shall see in the following section. However, GRAM has a slight edge in the DenseNet scenarios (e.g., CIFAR-100 vs. SVHN). We posit that the incremental residual connections of the DenseNet architecture (i.e., concatenation of input and output features) lead to a higher correlation between individual layers' test statistics (see Appendix Fig. 6). Ultimately, it leads to a Fisher test statistic with a higher variance while degrading the performance of MaSF (see discussion in Appendix D.2).

In contrast, Deep Mahalanobis and ResFlow perform exceptionally well in several scenarios but fail in others. In particular, CIFAR-100 (as in-dist) appears to be challenging in contrast to SVHN (as in-dist). Both methods rely on Mahalanobis distance which involves estimating and inverting large covariance matrices. Thus, we conjecture that the performance loss can be attributed to the limited number of samples in the dataset. This is a known phenomenon. For instance, Bai and Saranadasa (1996) showed a decrease in the Hotelling test power as the number of observations approaches the number of dimensions. We note that the authors use a Linear Discriminant Analysis (LDA) assumption (i.e., the covariance is identical across classes) to reduce the number of estimated parameters, however, it appears to be insufficient in this case.

Notably, no method consistently outperforms the others. It is expected since a uniformly most powerful test cannot be designed without access to the test distribution (Birnbaum, 1954). In essence, for each reasonable detection method, a setting exists for which it is most powerful. This fact highlights the importance of our framework as a principled approach for constructing OOD detectors.

Finally, we show that MaSF generalizes well to other scenarios through an extensive evaluation in Appendix G. Also, we demonstrate and analyze how to incorporate alternative test statistics from GRAM and Deep Mahalanobis within our framework (see Appendix G.2 and G.1).

## 4.4 EVALUATION OF COMPUTATIONAL COST

Computational cost is a key factor when choosing an algorithm for a specific application that was often overlooked by prior work. A direct comparison between different methods may not be trivial. This is due to the fundamental differences between procedures and the potential for optimizing specific implementations. Thus, we suggest measuring the Test statistic Computation Time (TCT) to approximate the cost of similar methods (i.e., methods that use a form of summary functions over intermediate feature maps). Table 3 presents a simple benchmark measuring the mean TCT (mTCT) and the mean global TCT time over all convolution layers in MobileNet-V2 (Sandler et al., 2019). This serves as a proxy for a typical CNN intended for edge devices, where resources are limited. In addition, we report the DNN's compute time (measured independently from TCT) as a reference.

Table 2: TPR at 95% of competing detectors on a popular OOD benchmark.

| Network | In-dist | Out-of-dist | MSP | Mahalanobis | ResFlow[a] | GRAM | MaSF(ours) |
|---|---|---|---|---|---|---|---|
| DenseNet | CIFAR-10 | SVHN | 40.3 | 89.6 | 86.1 | 96.1 | **98.4** |
| | | TinyImageNet | 59.4 | 94.9 | 96.1 | **98.8** | 97.8 |
| | | LSUN | 66.9 | 97.2 | 98.1 | **99.5** | 99.0 |
| | CIFAR-100 | SVHN | 26.3 | 62.2 | 48.9 | **89.3** | 83.7 |
| | | TinyImageNet | 17.5 | 87.2 | 91.5 | **95.7** | 93.9 |
| | | LSUN | 16.7 | 91.4 | 95.8 | **97.2** | **97.2** |
| | SVHN | CIFAR-10 | 61.8 | **97.5** | 90.0 | 80.4 | 86.8 |
| | | TinyImageNet | 80.5 | **99.9** | 99.9 | 99.1 | 99.8 |
| | | LSUN | 80.2 | **100** | 100.0 | 99.5 | 99.9 |
| ResNet | CIFAR-10 | SVHN | 27.8 | 75.8 | 91.0 | 97.6 | **99.0** |
| | | TinyImageNet | 42.3 | 95.5 | 98.0 | **98.7** | 98.4 |
| | | LSUN | 41.3 | 98.1 | 99.1 | 99.6 | **99.7** |
| | CIFAR-100 | SVHN | 15.1 | 41.9 | 74.1 | 80.8 | **89.7** |
| | | TinyImageNet | 17.7 | 70.3 | 77.5 | 94.8 | **96.1** |
| | | LSUN | 15 | 56.6 | 70.4 | 96.6 | **98.2** |
| | SVHN | CIFAR-10 | 79.2 | 94.1 | 96.6 | 85.8 | **98.0** |
| | | TinyImageNet | 74.7 | 99.2 | **99.9** | 99.3 | **99.9** |
| | | LSUN | 78.5 | 99.9 | **100.0** | 99.6 | **100.0** |
| | mTPR | | 46.7 | 86.1 | 89.6 | 94.9 | **96.4** |
| | SD | | 15 | 17.4 | 13.8 | 6.4 | **4.7** |
| | Min-TPR95 | | 25.9 | 41.9 | 48.9 | 80.4 | **83.7** |

[a]Results for Mahalanobis and ResFlow when tuned using adversarial examples and input pre-processing.

Results include Mahalanobis distance (under a LDA assumption) as a popular baseline, GRAM deviation score as the lead competitor, and our proposed method (MaSF). We find that MaSF mTCT is smaller by a factor of x35 and x2.5 when compared to the GRAM and the Mahalanobis statistics. Furthermore, Mahalanobis' TCT in Table 3 reflects a significantly lower cost compared to the full procedure from Lee et al. (2018) (as reported in Table 2), since it does not include the costly input pre-processing strategy. This strategy involves introducing perturbations to the input features at test time to increase their likelihood under the predicted class via backpropagation and a second forward pass. Moreover, the MaSF statistic does not involve general matrix-matrix multiplication. Hence, its computation can run concurrently on appropriate hardware without blocking the resources required for the CNN acceleration. Lastly, the abundant resources used to produce Table 3 are in favor of the alternative methods. Therefore the observed speedup of MaSF is expected to increase on small devices with limited resources. The full settings for this benchmark are in Appendix C.

## 4.5 RANDOM CHANNELS SELECTION

A popular trend in CNN design favors large models, where the number of parameters is greater than the number of available training samples. Naturally, the number of estimated parameters in the OOD detector is likely to increases as well. This presents a challenge from two aspects. First, from the statistical efficiency perspective, more samples are required to achieve the same power. Second, it leads to a substantial computational burden.

We explore dimensionality reduction via channel selection and focus on a simple scenario where a subset of channels is randomly selected for each layer. These subsets remain fixed during the measurement and evaluation phases of the test statistics. Table 4 presents the mTPR of the MaSF variants on the same benchmark as in Table 2: the channels are sampled uniformly for each layer, reducing the total number of channels by a given rate. Results are averaged over five random seeds.

Surprisingly, the MaSF statistic's power is not dramatically affected by the proportion of channels, indicating that the OOD signal is present across all channels. This implies that in some cases, channel sampling can be used to decrease the method test-time cost. In this case, MaSF suffers from negligible degradation ($\sim 1\%$) while using only 10% of the total number of channels, theoretically reducing the TCT cost by a factor of 10. In Appendix G.1, we show how sampling can be used to improve the Mahalanobis test statistic.

Table 3: Test statistic compute time.

| Method \ TCT | Single (ms) | | Total (ms) | | Relative[a] |
| --- | --- | --- | --- | --- | --- |
| | Mean | SD | Mean | SD | |
| MaSF (ours) | 0.22 | 0.03 | 11.6 | 0.23 | 0.93 |
| Mahalanobis[b] | 0.54 | 0.13 | 28.7 | 0.39 | 2.3 |
| GRAM | 7.56 | 0.93 | 393.6 | 42.0 | 31.6 |

[a] $\frac{\text{Total TCT}}{\text{DNN Time}}$, DNN (Forward) Time = $12.44 \pm 0.32$ ms.
[b] Without pre-processing required for Table 2 results, which adds backward and forward passes.

Table 4: MaSF with random channels.

| Channels | 5% | 10% | 25% | 50% | 75% | 100% |
| --- | --- | --- | --- | --- | --- | --- |
| Reduction[a] | x20 | x10 | x4 | x2 | x1.33 | x1 |
| mTPR | 94.5 | 95.2 | 95.8 | 96.1 | 96.3 | 96.4 |
| SD | 6.7 | 6.1 | 5.5 | 5.14 | 4.8 | 4.7 |
| Min-TPR95 | 78.5 | 80.8 | 81.4 | 82.4 | 83.7 | 83.7 |

[a] Theoretical cost reduction for MaSF TCT is linear in the number of channels.

## 5  RELATED WORK

In the following section, we review the relevant observer methods and related work.

Hendrycks and Gimpel (2016) proposed using the Maximum Softmax Probability (MSP) as a baseline method for OOD detection, positioned on the observation that a well-trained DNN tends to assign a higher probability to in-distribution vs. OOD examples. Liang et al. (2017) presented ODIN as an improvement to the baseline method by combining Softmax temperature scaling with a costly input pre-processing strategy. The temperature and perturbations magnitude hyper-parameters are calibrated on OOD samples. Lee et al. (2018) suggested the Deep Mahalanobis detector employing Mahalanobis distance over per-channel mean values to incorporate features from deep layers. At each layer, the minimal distance is selected over all classes. The results are combined using a weighted average, where the weights are determined via logistic regression on a small subset of the OOD dataset. The authors suggested replacing OOD data with adversarial examples as a proxy with diminished results. Zisselman and Tamar (2020) extends the Mahalanobis detector with a learned likelihood function, using deep residual-flow models (ResFlow). Similarly to Lee et al. (2018), the method leverages the aforementioned pre-processing strategy, and the per layer scores are also combined weighted average following the same procedures. Later, Sastry and Oore (2019) (GRAM) suggested a score based on the outer product (Gram matrix) over intermediate feature maps. The authors consider the total deviation from the minimal and maximal values observed on the training data for several Gram matrix exponent orders. Each layer contribution to the total sum is normalized by the mean deviation (of the predicted class) over a portion of the validation set. Recently, Liu et al. (2020) suggested a mutator method that employs an energy-score that is maximized for OOD samples during training (similar to Hendrycks et al. (2018)). This score computation is efficient and can be used without tuning the model. However, its results deteriorate compared to SOTA observer methods.

To the best of our knowledge, Grosse et al. (2017) were the first to adopt a statistical hypothesis test to detect adversarial examples in DNNs. The proposed method is based on the Maximum Mean Discrepancy (MMD) test for equality of distributions between two groups. The test statistic distribution is estimated using permutations (i.e., shuffling the labels of the two compared groups). However, the method can only be used on groups with a significant number of OOD samples. Hence, it cannot flag a single observation as OOD while it also requires extensive computing resources. Later, Sun and Lampert (2019) suggested utilizing the Kolmogorov-Smirnov test to compare the distribution of the observed percentiles of the model's Softmax scores with the uniform distribution. The test can efficiently determine if a network operates outside its specification (e.g., processed batch includes OOD samples). However, it is not suited for a single sample granularity.

A parallel line of work by Papernot and McDaniel (2018) proposed a method for robust prediction by measuring the disagreement between a test sample and its K-Nearest Neighbors (K-NN) within the training data (i.e., the number of samples whose label is different from the candidate label). The proposed algorithm adds the disagreement counts from all layers. It compares the result to the empirical distribution (measured using a holdout set as in Papadopoulos (2008)), providing a confidence measure that can be used for OOD detection. Recently, Raghuram et al. (2020) suggested an approach based on statistical hypothesis testing for detecting adversarial attacks. The proposed test statistics are also based on K-NN. The method obtains $p$-values per layer then combines them using classical combinations tests. However, the procedure's ultimate output is not a valid $p$-value,

as the dependency between the combined $p$-values is ignored (i.e., the assumption regarding the combined test statistic distribution does not hold). Moreover, a K-NN based approach has two crucial drawbacks. First, K-NN requires computing the features' distance per layer between the test sample and the entire training dataset at test time. Second, due to the use of full feature maps, this approach suffers from the phenomenon known as "the curse of dimensionality" when used in conjunction with large inputs. These flaws can be associated with Raghuram et al. (2020) performance on simple OOD benchmarks compared to Lee et al. (2018).

Finally, model calibration/uncertainty-estimation is a related topic that was a subject of extensive study (Zadrozny and Elkan, 2001; 2002; Platt and Karampatziakis, 2007; Naeini et al., 2015; Guo et al., 2017). The domain's objective is to assign confidence scores that correspond with the probability of the model error (e.g. by applying temperature scaling before SoftMax (Guo et al., 2017)). It defers from OOD detection since the test and training samples are drawn from similar distributions, and a correct prediction exists. Therefore, datasets, benchmarks, and metrics designed for one space are generally irrelevant to the other.

## 6 DISCUSSION

This paper presents a novel framework for OOD detection in DNNs based on statistical hypothesis testing. Our approach does not rely on prior knowledge regarding the test distribution nor changing the DNN's parameters. Next, we present a detection scheme dubbed MaSF (Max-Simes-Fisher). MaSF is compared to current OOD detection methods and demonstrates equivalent or better detection accuracy. Our procedure is also more efficient compared to other methods, a crucial property for real-world applications.

We suggest that without any assumptions regarding the test distribution, a uniformly most powerful test for OOD detection cannot be constructed (i.e., "no free lunch", Birnbaum (1954)). Therefore, a reasonable way to deal with OOD in the wild is to maintain a set of specialized detectors that are updated for new OOD patterns as they emerge. Such detectors can be constructed and combined within our framework. Furthermore, modern combination methods can be used to improve the detection accuracy. For example, tests such as Vovk and Wang (2020) can deal with dependent $p$-values, or tests using random sampling of features as in Frostig and Benjamini (2021).

While our framework is described in the context of classification, it can easily apply for regression (similarly to how the intermediate layers' features are handled). Class-dependent statistics could be used by assigning meta-classes to groups of inputs (e.g., via clustering) or by using a one-class approach. We intend to explore this in future work.

Moreover, the proposed framework can be used for other applications as well. One potential avenue is to test various hypotheses on the DNN itself. For instance, one can detect which channels are significant for the detection of specific classes using methods such as Benjamini and Bogomolov (2014); Heller et al. (2018). This can be used for network analysis or even to reduce the computational cost of inference. Other use cases include active or continual learning as a scoring mechanism to detect novel samples. Another example is to use our framework to filter unwanted samples (i.e., outliers) from large unlabeled datasets in self/semi-supervised scenarios.

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

# A    NOTATION

$\chi^c$      The set of all images of class $c$ in the measurement set

$\hat{\mathbb{P}}(x; g, \chi^c)$ Empirical distribution of function $g$ for class $c$, $\hat{\mathbb{P}}(x; g, \chi^c) = \frac{1 + \sum_{(X_i, y_i) \in \chi^c} I(t_{j,l}(X_i) \leq x)}{(n_c + 1)}$.

$\mathcal{P}^c$      Class $c$ image distribution

$a_l$      The number of channel at layer $l$

$F$      The convlutional neural network

$q^c(X)$      Class-conditional $p$-value of image $X$, given class $c$ [5].

$q_{j,l}^c(X)$    Class-conditional $p$-value of channel $j$ in layer $l$ for image $X$, given class $c$.

$q_{\max}$      The maximum $p$-value for an image across all classes, $q^{\max}(X) = \max\{q^1(X), \ldots, q^k(X)\}$

$t^c$      Layer reduction function, $t^c(X_i) = T^L(t_1^c(X_i), \ldots, t_L^c(X_i))$

$T^L$      The layer reduction function

$T^s$      Spatial dimensions reduction function

$T^{ch}$      Channel reduction function

$T_{\text{Fisher}}(\boldsymbol{q})$ Fisher test statistic, $T_{\text{Fisher}}(\boldsymbol{q}) = -2 \sum_{i=1}^{m} \ln(q_i)$

$T_{\text{Simes}}(\boldsymbol{q})$ Simes test statistic, $T_{\text{Simes}}(\boldsymbol{q}) = \min\limits_{i \in \{1, \ldots, m\}} q_{(i)} \frac{m}{i}$

$t_{j,l}$      Spatial reduction function value at channel $j$ in layer $l$, $t_{j,l}^c(X_i) = T^S(F_{j,l}(X))$

$t_l^c$      Channel reduction function value at layer $l$, $t_l^c(X_i) = T^{ch}(q_{1,l}^c(X_i), \ldots, q_{a_l,l}^c(X_i))$

$X_i$      Image $i$

$y_i$      The class of image $i$

# B    STATISTICAL PROCEDURE DETAILS

## B.1    THEORETICAL VALIDITY

We are interested in two distinct hypotheses to test. The first one, is to identify if the image is OOD,

$$\mathcal{H}_0^* : \exists c : X_{\text{test}} \sim \mathcal{P}^c, \quad \mathcal{H}_1^* : X_{\text{test}} \not\sim \mathcal{P}^c \quad \forall c \in [k], \tag{7}$$

i.e., the goal is to find if there does not exist a class distribution which the image was sampled from. The null hypothesis can be tested using $q^{\max}(X_{\text{test}})$, yielding a valid test (i.e., a test which maintains the significance level, $\alpha$).

The second hypothesis of interest is $\mathcal{H}_0^{\hat{y}_{\text{test}}}$,

$$\mathcal{H}_0^{\hat{y}_{\text{test}}} : X_{\text{test}} \sim \mathcal{P}^{\hat{y}_{\text{test}}}, \quad \mathcal{H}_1^{\hat{y}_{\text{test}}} : X_{\text{test}} \not\sim \mathcal{P}^{\hat{y}_{\text{test}}}, \tag{8}$$

In this case, we are interested to know if the image is sampled from the same distribution of the class predicted for it.

When proving Proposition 1, we will rely on the fact that the maximum p-value is always larger or equal to the true-class p-value. In Proposition 2, we will show that when the class-conditional p-value is used, the accuracy of the model needs to be taken into account.

Important for both proofs, is that for sample $X, y$, the distribution of the test statistic $q^y(X)$ is stochastically at least as large as the uniform distribution (it is in fact larger than uniform since the test statistic is discrete) under the null hypothesis, that is, $P_{H_0}(q^y(X) \leq \alpha) \leq \alpha$ (see Vovk et al. (2005) Proposition 4.1).

The assumptions required for our propositions are the following:

---

[5]For right sided tests, $q^c(X) = 1 - \hat{\mathbb{P}}^c(X)$.

1. For each class $c$, the images in $(X_i, y_i) \in \chi_{\text{val}}^c$ are independent identically distributed.
2. $\chi_{\text{train}}$ is independent of $\chi_{\text{val}}$.
3. $(y_{\text{test}}, X_{\text{test}})$ are independent of $\chi_{\text{val}}$.

**Proof of Proposition 1** We need to show that $q^{max}(X_{test})$ is stochastically at least as large as the uniform distribution when $\mathcal{H}_0^*$ is true, i.e., $y_{\text{test}} \in [k]$. Since $q^{y_{\text{test}}}(X_{test})$ is a valid $p$-value for $\mathcal{H}_0^{y_{test}}$ (Vovk et al. 2005 Proposition 4.1), it follows that

$$P_{\mathcal{H}_0^{y_{test}}}(q^{y_{\text{test}}}(X_{test}) \leq \alpha \mid \chi_{train}) \leq \alpha. \tag{9}$$

Inequality (9) follows since $t^c(X_{test}; \chi_{train})$ is exchangeable with $\{t^c(X; \chi_{train}) : X \in \chi_{val}\}$. Since $q^{\max}(X_{\text{test}}) \geq q^{y_{\text{test}}}(X_{\text{test}})$, then

$$
\begin{aligned}
P_{\mathcal{H}_0^*}(q^{\max}(X_{\text{test}}) \leq \alpha \mid \chi_{train}) &\leq P_{\mathcal{H}_0^*}(q^{y_{\text{test}}}(X_{\text{test}}) \leq \alpha \mid \chi_{train}) \\
&= P_{\mathcal{H}_0^{y_{test}}}(q^{y_{\text{test}}}(X_{\text{test}}) \leq \alpha \mid \chi_{train}) \\
&\leq \alpha,
\end{aligned}
\tag{10}
$$

concluding the proof.

We now turn to find the required adjustment presented in Proposition 2.

**Proof of Proposition 2** In the following proof we will bound the probability of T1E as a function of the classifier accuracy.

$$
\begin{aligned}
P_{\mathcal{H}_0^{y_{test}}}(q^{\hat{y}_{\text{test}}}(X_{test}) \leq \alpha) &= P_{\mathcal{H}_0^{y_{test}}}(q^{\hat{y}_{\text{test}}}(X_{test}) \leq \alpha | \hat{y}_{\text{test}} = y_{\text{test}}) \times P(\hat{y}_{\text{test}} = y_{\text{test}}) + \\
&\quad P_{\mathcal{H}_0^{y_{test}}}(q^{\hat{y}_{\text{test}}}(X_{test}) \leq \alpha | \hat{y}_{\text{test}} \neq y_{\text{test}}) \times P(\hat{y}_{\text{test}} \neq y_{\text{test}}).
\end{aligned}
$$

The equality is according to the law of total probability. Since $\hat{y}_{test}$ is only a function of $\chi_{train}$ (and $X_{\text{test}}$, it follows that

$$P_{\mathcal{H}_0^{y_{test}}}(q^{\hat{y}_{\text{test}}}(X_{test}) \leq \alpha | \hat{y}_{\text{test}} = y_{\text{test}}) = E(P_{\mathcal{H}_0^{y_{test}}}(q^{y_{\text{test}}}(X_{test}) \leq \alpha | \chi_{train}) \mid \hat{y}_{\text{test}} = y_{\text{test}}) \leq \alpha,$$

where the expectation is over the distribution of $\chi_{train}$ conditional on the event $\hat{y}_{\text{test}} = y_{\text{test}}$, and the inequality follows from (9).

Moreover, $P_{\mathcal{H}_0^{y_{test}}}(q^{\hat{y}_{\text{test}}} \leq \alpha | \hat{y}_{\text{test}} \neq y_{\text{test}}) \leq 1$. Therefore,

$$P_{\mathcal{H}_0^{y_{test}}}(q^{\hat{y}_{\text{test}}}(X_{test}) \leq \alpha) \leq \alpha \times P(\hat{y}_{\text{test}} = y_{\text{test}}) + 1 \times P(\hat{y}_{\text{test}} \neq y_{\text{test}}),$$

concluding the proof.

Assuming $\hat{y}_{\text{test}}$ is given by the CNN classification, then if the network does not make any mistake, the test is valid. If not, then the significance level needs to be adjusted. (For our experiments no adjustment of the significance level is necessary, since we already calibrate the significance level in our experiments in order to compare with the various competing methods, as detailed in Section 4.1.)

Note, that the conformal p-values are obtained by conditioning solely on the training set, resulting p-values that are dependent on the validation set (T1E is guaranteed in expectation over all existing test samples and validation sets). Alternatively, Bates et al. (2021) suggested a method in which the T1E guarantee is obtained conditionally on the validation set as well. It ensures that the resulting p-values are independent but comes at a cost of decreased power. In practice, we find the conformal $p$-values are conservative implying the T1E is maintained (even on a specific validation set, see next section).

## B.2 EMPIRICAL EVIDENCE OF METHOD VALIDITY

This section presents simulation and methods relating to the validity of the suggested framework and MaSF specifically. According to the theory presented above, the resulting $p$-values from our

procedure should follow a uniform (or stochastically larger than uniform) distribution. We graphically assess the resemblance of the distributions using a qq-plot, in which the quantiles of the uniform distribution are plotted vs. the quantiles of the $p$-values distribution. The identity line will represent a distribution exactly matching the uniform distribution quantiles. Curves below/above it represent stochstically smaller/larger than uniform distributions.

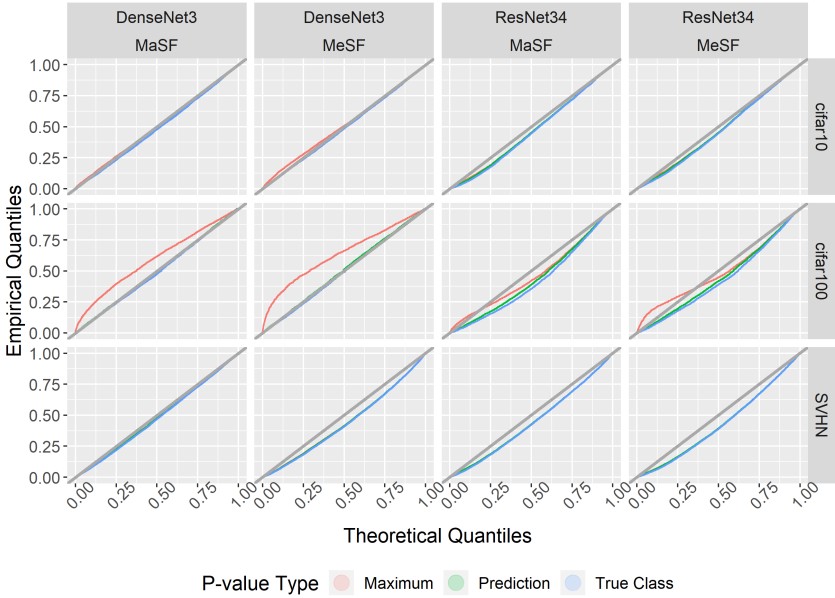

Figure 1: $p$-values from MaSF and MeSF comparison to uniform [0,1] distribution. eCDFs are estimated using the training data.

We begin by examining the max $p$-values when using Lee et al. (2018) benchmark, used to report Table 2 results. The distribution indeed appears uniform or stochastically larger, e.g. the maximum (although there are no theoretical guarantees for it). The exception is ResNet-34 in which, when inspecting quantiles away from the tails, the distribution appear stochastically smaller than uniform.

In order to assess our theoretical guarantees of the framework we abandon Lee et al. (2018) benchmark. We split our data into three sets: 1. DNN training set, $\chi_{\text{train}}$ 2. OOD validation set, $\chi_{\text{val}}$ 3. Test set. The eCDFs of the spatial and channel reduction is estimated using the training-set of the DNN. This introduces bias to the eCDF estimation caused by the dependency between the DNN weights and the eCDFs. The alternative, is to use a different independent set of observations, wasting valuable resources. Note, this does not harm the procedure validity. We the use set $\chi_{\text{val}}$ to estimate the final layer reduction eCDF and $\chi_{\text{train}}$ in order to estimate the spatial and channel reductions eCDF. Since, $\chi_{\text{val}}$ is independent of $\chi_{\text{train}}$ it ensures that we obtain independent $p$-values, thus our proposition hold. Intuitively, if the $p$-values at the layer level are obtained from the same data that is used to estimate the channel reduction eCDF, they will tend to be larger than $p$-value resulting from observations independent of the eCDF. This makes new observations to appear "unusual".

We examine the resulting $p$-values using this procedure on the test test, and they are assessed using qq-plots (Fig. 1, 2). In our experiments we focused on SVHN and CIFAR-10 (CIFAR-100 has a smaller validation set). We split their validation set as follows (the number of samples is per class): the DNN training-set is used for the channels reduction eCDF, 800 samples for estimating the layers reduction eCDF, and 200 samples to inspect the $p$-values distribution.

We use the the simulation to demonstrate the theory. The $p$-values indeed follow a uniform distribution, when splitting the data and ensuring the independence of each step. This implies that for all significance levels our framework yields a valid test. The subsequent split of the validation set is required in all methods in which the test statistic is a function of estimated distributions. In Table 5 it can be seen that the power of the valid method does not decrease substantially compared to the experiment version. In both cases it seems that the MaSF variant is the most powerful.

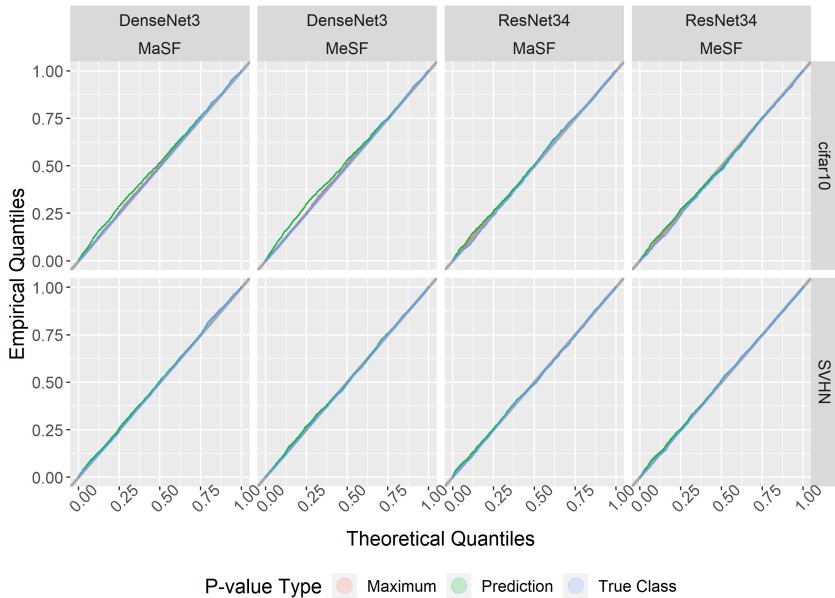

Figure 2: $p$-values from MaSF and MeSF comparison to uniform [0,1] distribution. eCDFs are estimated on holdout data from the validation set. CIFAR-100 was removed due to the small number of observations.

Table 5: Comparison of the suggested framework various OOD detectors. F - Fisher, S - Simes, Me - Mean and Ma - Max, the order is first the spatial reduciton (Me / Ma), channel reduction (F / S) and finally layer reduction (F / S).

| Network | In-dist | Out-of-dist | TPR at 95% | | | | AUROC | | | |
| --- | --- | --- | --- | --- | --- | --- | --- | --- | --- | --- |
| | | | MaFF | MaSF | MeFF | MeSF | MaFF | MaSF | MeFF | MeSF |
| DenseNet | CIFAR-10 | SVHN | 98.3 | **98.4** | 94.5 | 95.3 | 98.4 | **98.6** | 94.7 | 96.1 |
| | | TinyImageNet | 95 | **97.7** | 88.4 | 91.2 | 93.3 | **97.7** | 88.4 | 91.3 |
| | | LSUN | 97.4 | **99** | 93.2 | 95.2 | 96.3 | **99.1** | 93.5 | 95.7 |
| | SVHN | CIFAR-10 | 85.5 | **87.1** | 62.1 | 61.6 | 91 | **93.2** | 84.5 | 83.5 |
| | | TinyImageNet | 99.6 | **99.8** | 92.6 | 93.7 | 99.8 | **99.9** | 97.9 | 98.6 |
| | | LSUN | **99.9** | **99.9** | 90.1 | 91.5 | **100** | **100** | 98 | 98.7 |
| ResNet34 | CIFAR-10 | SVHN | **99.2** | 99 | 97.2 | 97.4 | **98.8** | 98.4 | 96.9 | 96.3 |
| | | TinyImageNet | 96.5 | **98.5** | 93.3 | 94.3 | 94 | **97.4** | 91.6 | 93.7 |
| | | LSUN | 99.1 | **99.7** | 98 | 98.4 | 98.1 | **99.3** | 97 | 97.9 |
| | SVHN | CIFAR-10 | 97.5 | **98** | 95.2 | 93.9 | 95.2 | **96** | 95.4 | 95 |
| | | TinyImageNet | 99.8 | **99.9** | 99.3 | 99.3 | 99.7 | **99.8** | 99.4 | 99.3 |
| | | LSUN | 99.8 | **100** | 98.9 | 98.9 | 99.8 | **100** | 99.4 | 99.3 |
| | Avg | | 97.3 | **98.1** | 91.9 | 92.6 | 97 | **98.3** | 94.7 | 95.4 |
| | SD | | 4 | **3.5** | 10 | 10.1 | 3 | **2** | 4.6 | 4.5 |
| | Min | | 85.5 | **87.1** | 62.1 | 61.6 | 91 | **93.2** | 84.5 | 83.5 |

## B.3 GLOBAL NULL TESTS

In some cases, we care about a set of hypotheses and testing whether none of these hypotheses are false (the global null). Combination tests are used in such cases. Let $\mathbf{q}$ denote a vector of $m$ $p$-values, $\boldsymbol{q} = (q_1, \ldots, q_m)$, used to test $m$ null-hypotheses, $H_{0,1}, \ldots, H_{0,m}$. The global null hypothesis is defined as $H_{0,.} = \bigcap_{i=1}^{m} H_{0,i}$. The test statistic is $T(\mathbf{q})$. Every combination test summarizes $\boldsymbol{q}$ differently. Hence their respective powers differ, depending on the unknown distribution of the test statistic under $\mathcal{H}_1$.

Two common tests of the global-null are Fisher (Fisher, 1992) and Simes (Simes, 1986). The Fisher combination test statistic is $T_{\text{Fisher}}(\boldsymbol{q}) = -2 \sum_{i=1}^{m} \ln(q_i)$. If the $p$-values are independent and have a uniform null distribution, then $T_{\text{Fisher}}(\boldsymbol{q}) \sim \chi^2_{2m}$ when the global null hypothesis is true. Otherwise, the null distribution of the test statistic is unknown, and its distribution needs to be estimated. For

cases, when not all hypotheses are of the same importance, the procedure can be modified to use weights, $T_{\mathrm{W-Fisher}}(\boldsymbol{q}; w) = -\sum_{i=1}^{m} w_i \ln(q_i),\ \sum_{i=1}^{m} w_i = 1$.

We can turn to permutation-testing to test the weighted version of the Fisher combination test or the standard version when the assumptions do not apply. We will estimate the CDF of the test statistic under the null-hypothesis. When a new observation arrives, one uses this eCDF to test the null-hypothesis.

Simes test (Simes, 1986), involves ordering the $p$-values, $q_{(1)} \le q_{(2)} \le \cdots \le q_{(m)}$, and calculating the following test statistic $T_{\mathrm{Simes}}(\boldsymbol{q}) = \min\limits_{i \in \{1,\ldots,m\}} q_{(i)} \frac{m}{i}$. If the $p$-values are independent and have a uniform null distribution, then $T_{\mathrm{Simes}}(\boldsymbol{q}) \sim \mathrm{Uniform}[0,1]$ when the global null is true, so rejecting the global null for $T_{\mathrm{Simes}} \le \alpha$ will maintain the T1E at level $\alpha$ (Simes, 1986). When these assumptions are not met, one may use the eCDF, as with the Fisher test statistic.

To demonstrate how these tests differ from one another, consider testing if $\mu$, the mean vector of $m$ features, is zero. The evidence against the null (signal) is dense if the number of non-zero entries in $\mu$ is large and sparse if only a few entries in $\mu$ are non-zero. The Simes test is best suited to detect a sparse signal, as small $p$-values will dominate the value of $T_{\mathrm{Simes}}$. In contrast, the Fisher combination test performs better when the signal is dense since $T_{\mathrm{Fisher}}$ aggregates the $p$-values of all features. See Cheng and Sheng (2017) for a simulation comparing the two methods.

## C  EXPERIMENTAL SETTINGS

**Models and in-dist datasets.** In our experiments we focus on popular vision architectures: DenseNet (Huang et al., 2016) and ResNet-v1 (He et al., 2016). We follow the benchmark proposed by Lee et al. (2018), reusing the same pretrained models, datasets and evaluation code to report our results. Each architecture (DenseNet-BC & ResNet-34) is paired with a set of weights, trained on CIFAR-10, CIFAR-100 (Krizhevsky et al., 2009) and SVHN (Netzer et al., 2011). the appropriate training dataset is referred to as the in-distribution while results are reported on the full validation split.

**OOD datasets.** We evaluate the OOD detection on the resized variants of LSUN (Burda et al., 2018) and Tiny-ImageNet (Wu et al., 2017) as processed and shared by Liang et al. (2017). We similarly include CIFAR-10 and SVHN validation splits when they are not used for model training.

**General calibration settings.** Our proposed method requires estimating the statistics' class conditional distributions (i.e., the empirical CDF) over the calibration set to extract $p$-values. Thus, we effectively collect a set of percentiles for each statistic and every class in the in-distribution dataset. In our experiments, each sample is used once and without any typical train-time augmentations (i.e., we only apply the required manipulations for inference such as normalization and resize).

Furthermore, since our objective is rejecting $\mathcal{H}_0^c$ or $\mathcal{H}_0^*$, we can focus on estimating percentiles at the edge of the spectrum for each statistic (i.e., the distribution tails), which improves the sensitivity to abnormal observations. For instance, when performing the Simes combination test, the output $p$-value is scaled proportionally to the number of channels and rank. Therefore, layers with many channels will require an extreme observation in one channel to reflect abnormality for the entire layer (see Appendix E for more details).

In our experiments, we estimate percentiles between $0.1$ and $0.9$ at $0.1$ increments. In addition, we collect smaller than $0.025$ and greater than $0.975$ for the two-tail test per-channel statistics. For the layer combination test, we simply use a uniform $p$-value resolution of 1e-3. We remind the reader that a high resolution of percentiles is not needed for testing but improves the estimation of the produced $p$-value distribution, which we use for the empirical validation of the method in Appendix B.2.

Since any hardware has a limited amount of memory, we cannot always observe the entire calibration data simultaneously. For simplicity, we average percentiles over fixed size batches. Therefore, the minimal percentile resolution is $\frac{1}{\text{batch size}}$. Our experiments use a batch size of 1000 samples of each class for models trained on CIFAR-10 and SVHN. For CIFAR-100, we use a batch-size of 500 due to the limited number of examples per class.

When the calibration set size permits (i.e., more samples can fit in a single batch), we also collect the top and bottom 200 values observed during the entire calibration process to improve tails quantiles

estimation. After observing the entire calibration set, we select 10 percentiles at regular intervals from the end of each tail. The resulting percentiles are determined by the total number of examples in the calibration data.

**Test statistic Computation Time.** Our measurements for Table 3 include:

- MaSF includes spatial max-pooling operation, followed by a sort operation required by the Simes test. MaSF $p$-value lookup time is discounted.
- Mahalanobis time includes average-pooling followed by Mahalanobis distance for all classes under LDA assumption. Computation is done in matrix form to optimize execution time. Moreover, it does not include input pre-processing which was used by the original authors to produce results reported in Table2.
- Gram time includes computation of the Gram deviation score over 10 matrix powers.
- Total inference time for MobileNet-V2 with 1K classes and a synthetic input of shape (1,3,224,224).

When calculating the mean of single and total (i.e., time per-sample) statistic execution time (TCT), we measure the wall time over the all convolution layers' outputs, that are induced using a synthetic input. Global-mTCT times are measured over 10k + 1k warm-up iterations, while statistics are computed sequentially (blocking next layer statistic until current statistic compute is done). Single-mTCT is also averaged over all monitored layers.

Hardware used to produce results is based on 2 x Intel(R) Xeon(R) Gold 6152 CPU @ 2.10GHz system with a 2080ti GPU. Environment is based on Ubuntu 18.04, PyTorch 1.6, cuda 10.2, cudnn 7.6.5. Network compute and TCT are decoupled and do not compete over compute resources.

# D    ADDITIONAL ANALYSIS

## D.1    CHANNEL REDUCTION STATISTIC DISTRIBUTION

In Fig. 3, we illustrate the distribution of mean and max spatial reductions per-channel in ResNet34 for a single class compared to all other classes within the calibration set.

The distributions appears to be uni-modal, while abnormality can be higher or lower than the target class statistic. This supports our choice of a two-tail test for MaSF. Similar assumption can be easily validated for any reduction and data during the design phase of a new method, given knowledge on the in-distribution.

We note that we did not find substantial evidence that favours prioritizing channels based on such inter-class discrimination, compared to a simple random choice in our preliminary OOD detection experiments. However, this is an intriguing avenue for future work.

## D.2    LAYER CORRELATION ANALYSIS

The correlation between the test statistics of different layers seems to be a property of the network rather than that of a specific dataset or the test statistic used (comparing Fig. 4 A and Fig. 6 B). It can be seen that for both ResNet34 and DenseNet, sequential as well as skip connection layers are strongly correlated and that the correlation tends to intensify as among layers at the end of the network (Fig. 4 - 6). Notably, the DenseNet correlation occurs even for distant layers, while in ResNet34 they tend to lessen rapidly as the layers are further apart. In addition, the correlation does vary greatly for different test statistics (see Fig. 5).

Our method uses the Fisher combination test to combine $p$-values from layers. Usually, it requires the assumption of independence, however, we have circumvented it by estimating the statistic eCDF (under $\mathcal{H}_0$). We find that when combining the log $p$-values the variance of the statistic can be very large due to the strong correlations. Since the power of the test is a function of the overlapping between the distributions under the null and alternative hypotheses, a reasonable strategy is to reduce the variance of the statistics.

A simple method of achieving the goal is to select certain layers. However, this can lead to bias since the signal we care for can be located in the omitted layers. Therefore, we use hierarchical clustering

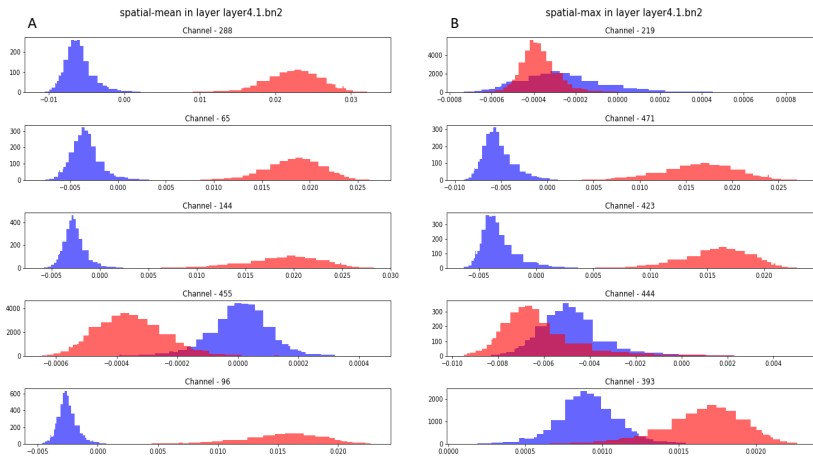

Figure 3: Histogram visualization of max and mean spatial reductions in the ResNet34 for selected layers and channels for two in-distribution classes of CIFAR-10.

to group together layers with low correlation. Then, at each cluster, we combine the $p$-values using the Fisher combination test. The final $p$-value is obtained using a simple Simes test. We found that in our experiments reducing the variance led to marginally improving the results, therefore they are not presented. We leave further study of this topic for future work.

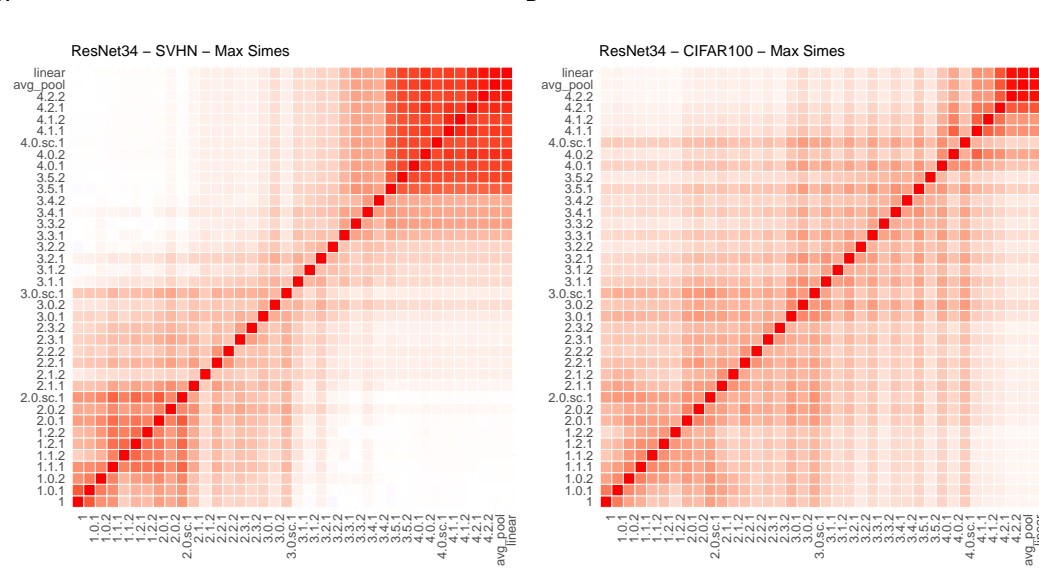

Figure 4: Comparing the correlation of layer level $p$-values for MaSF, between CIFAR-100 and SVHN with ResNet34. Red indicate positive correlation, and white neutral correlation.

# E    TESTING LAYERS WITH A LARGE NUMBER OF CHANNELS

We have used the Simes test to combine all channels' $p$-values to represent the layer. A possible alternative one may consider, is to simply choose the minimum $p$-value instead. The issue with this approach is its ignorance of the layer size (i.e., the number of channels). Suppose there are $n$ observations in our calibration set and $p$ channels, assuming the channels are independent, the

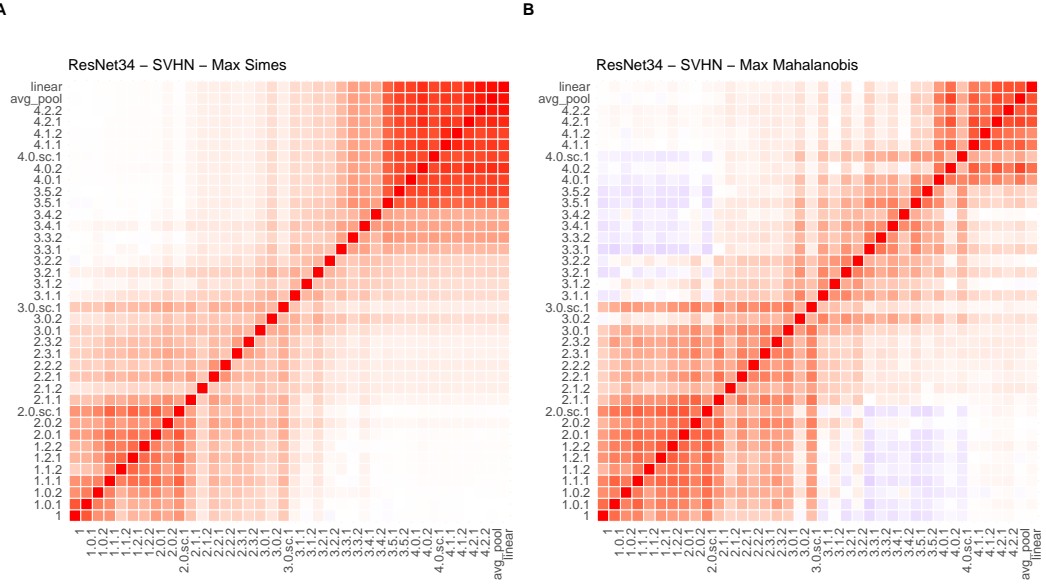

Figure 5: Comparing the Max-Simes statistic correlation with the Max-Mahalannobis statistic across layers for ResNet34 and SVHN. Red indicate positive correlation, and blue negative correlation.

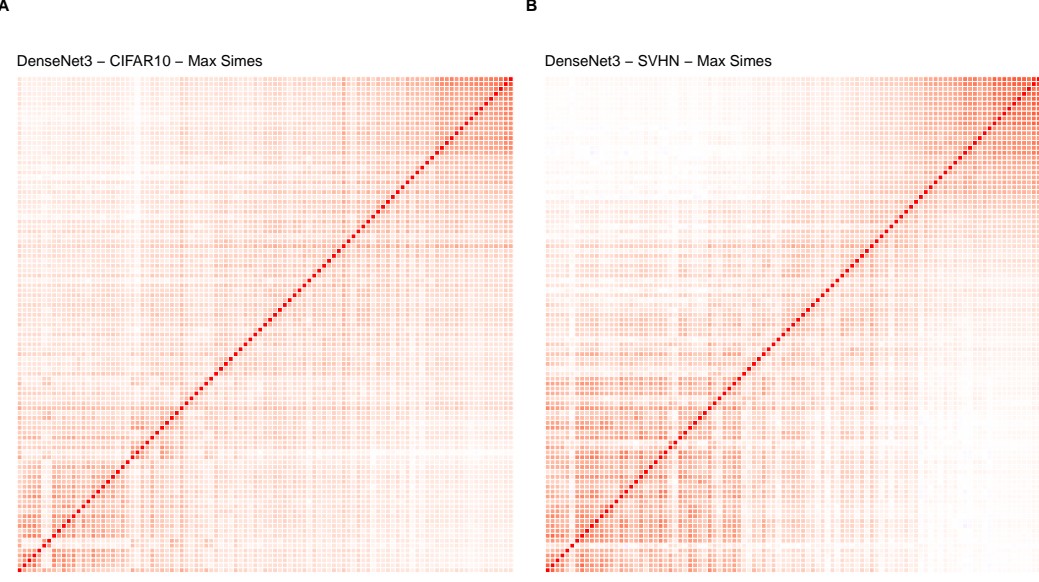

Figure 6: Comparing Max-Simes statistic across layers on CIFAR100 and SVHN for DenseNet3 model. Layer names are removed for eligibility. Red indicate positive correlation, and white neutral correlation.

probability of obtaining the minimal possible $p$-value, $1/n$, under the null hypothesis, is $(n - 1/n)^p$. As $p$ increase this probability tends to 1, rendering the layer uninformative. Thus, a good combination method should account for the total number of channels.

The problem is addressed by the Bonferroni correction, which is a common method for multiple comparisons corrections. It involves multiplying the channels' $p$-values by the number of hypotheses considered. Now, we can simply use the corrected minimal $p$-value. However, for a layer that has more channel than the number of available observations, $n < p$, the obtained $p$-value will always be $1$. This is due to the conservative correction rule. Since, the $p$-values obtained by the Bonferroni correction, are equal or larger than those from Simes test. The Simes test is uniformly more powerful than the Bonferroni test (Simes, 1986). Other multiple hypotheses correction methods could be applicable such as Hochberg (1988) and more. One still needs to choose the how summarize the $p$-values into a single value, for example taking the minimum corrected $p$-value. We have chosen the Simes test, since it is a powerful and simple test well suited for detecting sparse signal, with added benefit of adjusting for the number of channels.

A different option is to simply use any reduction and simply evaluate the eCDF of the reduction again. However, as was demonstrated with using the minimum these functions can be uninformative.

## F MAX-SIMES-FISHER ALGORITHM

Algorithm-2 describes the MaSF detector. It assumes that the channel reduction function $T^{ch}$ is the Simes test. Therefore, $t_l^c(X_{test}) \in (0,1)$, removing the need for another calibration. In the following sections we will experiment and analyze additional schemes by replacing the spatial and channel reductions.

---

**Algorithm 2:** MaSF: class-conditional $p$-values

**Input** : $F, X_{\text{test}}, c$ ;                // The network, input image and class of interest
**Input** : $\hat{\mathbb{P}}(.; t_{j,l}^c, \chi_{\text{train}}^c)\ j \in [a_l], l \in [L]$ ;   // eCDF per-channel after max spatial reduction
**Input** : $\hat{\mathbb{P}}(.; t^c, \chi_{\text{val}}^c)$ ;                          // eCDF of the Fisher test statistic
**Output**: $q^c(X_{\text{test}})$;                          // Class conditional $p$-value for the input image
**for** $l \in [L]$ **do**
    **for** $j \in [a_l]$ **do**
        $t_{j,l}(X_{\text{test}}) = \max(F_{j,l}(X_{\text{test}}))$
        $q_{j,l}^c = \min(\hat{\mathbb{P}}(t_{j,l}(X_{\text{test}}); t_{j,l}^c, \chi_{\text{train}}^c), 1 - \hat{\mathbb{P}}(t_{j,l}(X_{\text{test}}); t_{j,l}^c, \chi_{\text{train}}^c))$
    **end**
    Sort $\boldsymbol{q}, q_{(1)}, \ldots, q_{(a_l)}$
    $t_l^c(X_{test}) = \max_i(\frac{a_l}{i} q_{(i),l}^c(X_{\text{test}}))$ ;                          // Simes test
**end**

$t^c(X_{test}) = -2 \sum_{l=1}^{L} \log(t_l^c(X_{\text{test}}))$ ;                          // Fisher test
**Return** $1 - \hat{\mathbb{P}}(t^c(X_{\text{test}}); t^c, \chi_{\text{val}}^c))$

---

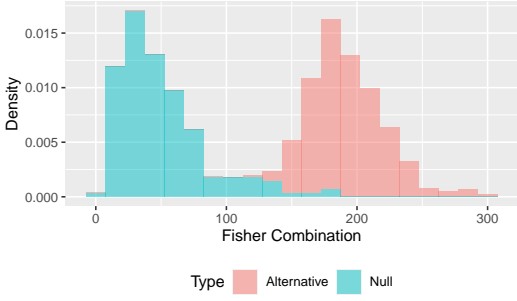

Figure 7: The empirical distribution of the class conditional MaSF test statistic for digit "0" from the SVHN dataset (blue). The alternative distribution is based on the remaining digits.

Table 6: Comparison of the adapted MeMF (Mean-Mahalanobis-Fisher) under LDA and GDA assumtions to Deep Mahalanobis (Lee et al., 2018). GDA suffers from the small number of samples in CIFAR100. Applying random channel selection significantly improves the detector. Deep Mahalanobis results include pre-processing. SDs are given in brackets where applicable.

| Network | In-dist | Out-of-dist | Mahalnobis | MeMF | MeMF (GDA) | MeMF @25% | MeMF @25% (GDA) |
|---|---|---|---|---|---|---|---|
| DenseNet3 | CIFAR-10 | SVHN | 88.7 | 90.1 | 83.1 | **92.5** (0.5) | 88.1 (0.6) |
| | | Imagenet | 88.6 | 94.7 | **96.6** | 93.1 (0.3) | 95.6 (0.2) |
| | | LSUN | 92.4 | 97.3 | **98.7** | 96.6 (0.2) | 98.3 (0.1) |
| | CIFAR-100 | SVHN | 48.7 | 65.8 | 47.6 | **71.0** (2.2) | 60.8 (0.3) |
| | | TinyImagenet | 80.4 | **89.8** | 83.0 | 87.1 (0.8) | 86.7 (0.6) |
| | | LSUN | 83.8 | **93.3** | 84.6 | 89.9 (1) | 87.6 (0.8) |
| | SVHN | CIFAR-10 | **92.5** | 78.5 | 87.1 | 76.2 (1.9) | 80.4 (0.9) |
| | | TinyImagenet | 99.1 | 99.1 | **99.6** | 98.6 (0.1) | 99.3 (0.1) |
| | | LSUN | 99.7 | 99.2 | **99.8** | 98.8 (0.3) | 99.5 (0.1) |
| ResNet34 | CIFAR-10 | SVHN | **87.5** | 67.4 | 64.8 | 84.1 (0.6) | 81.7 (0.5) |
| | | TinyImagenet | 93.1 | 92.7 | 93.7 | 93.6 (0.3) | **94.8** (0.2) |
| | | LSUN | **99.9** | 95.9 | 97.9 | 97.0 (0.4) | 98.3 (0.2) |
| | CIFAR-100 | SVHN | **66.5** | 22.9 | 0.0 | 36.0 (1.7) | 41.5 (1) |
| | | TinyImagenet | 56.7 | 87.6 | 0.0 | **88.4** (0.5) | 82.9 (0.4) |
| | | LSUN | 38.4 | **90.1** | 0.0 | 89.4 (0.7) | 82.8 (0.7) |
| | SVHN | CIFAR-10 | 95.2 | 97.1 | **98.6** | 96.0 (0.1) | 97.9 (0) |
| | | TinyImagenet | 99.3 | 99.7 | **99.8** | 99.7 (0.0) | 99.7 (0) |
| | | LSUN | 99.9 | **100.0** | 99.9 | 99.8 (0.0) | 99.9 (0) |
| | Average | | 83.9 | 86.7 | 74.2 | **88.2** | 87.5 |
| | SD | | 18.8 | 18.4 | 35.8 | **14.8** | 14.9 |
| | Min | | 38.4 | 22.9 | 0.0 | 36.0 | **41.5** |

# G ADDITIONAL EXPERIMENTS

## G.1 MAHALANOBIS EXAMPLE

In Table 6, we present the full results of adapting the Mahalanobis statistics to our framework, dubbed MeMF (Mean-Mahalanobis-Fisher). This is done by computing the Mahalanobis distance over the spatial means for all channels. The resulting distance is converted into a layer $p$-value using the appropriate eCDF. Finally, the layers are combined using the Fisher test statistic.

Our approach removes the necessity of OOD proxy for calibration, while the resulting detector is guaranteed to maintain T1E. We provide results with and without the LDA assumption (which we refer to as GDA). It can be seen that by utilizing our framework, we improve Mahalanobis detector results (LDA) by 2.82%, although it does not use costly input pre-processing. When LDA assumption is removed (GDA) the Mahalanobis detector severely degrades CIFAR-100 as expected due to the small sample size. We also evaluate random channel selection similarly to Section 4.5, results in Table 6 are for the best channel sampling rate. Applying random channel selection significantly improves the detector, specifically in previously failed scenarios.

## G.2 GRAM METHOD ANALYSIS

The GRAM procedure works well, as evident by the results. We wish to understand which elements are essential to its success. To do so, we segment the method into 3 components, considering a simple formulation based on a single Gram matrix power.

**Spatial reduction.** The Gram matrix is effectively used to reduce the spatial dimensions to a single value per-channel via row summation. Then, the min and max parameters are estimated using the observed values per-class. Finally, the statistic can be seen as the per-channel deviation from the estimated extreme parameters. We note that these parameters are essentially quantiles that depend on the total number of examples per class.

**Channel reduction.** Each layer is summarized using the sum over all per-channel deviations.

**Layer reduction.** Since the number of channels can impact the scale of each layer deviation. The layer deviation is divided by the expected deviation of the layer ($E_{va}(\delta_l)$), which is measured on 10% of the validation set. This normalizes the contribution of layers with a different number of channels

Table 7: Evaluating the importance of key components from GRAM method within our proposed framework. Replacing the Gram matrix with simple max pooling slightly improves OOD detection results. This indicates that the deviation function is more important than the Gram matrix which is the compute intensive part of the method.

| Network | In-dist | Out-of-dist | TPR at 95% | | |
| | | | $GP_1$ | $GP_1+\delta^*$ | $Max+\delta^*$ |
|---|---|---|---|---|---|
| DenseNet | CIFAR-10 | SVHN | 0.22 | **95.3** | 94.7 |
| | | TinyImageNet | 78.1 | 91.5 | **97.5** |
| | | LSUN | 83.3 | 95.9 | **99.0** |
| | CIFAR-100 | SVHN | 0.58 | **85.4** | 81.3 |
| | | TinyImageNet | 77.0 | 85.7 | **94.2** |
| | | LSUN | 81.7 | 87.0 | **97.0** |
| | SVHN | CIFAR-10 | 43.5 | 72.6 | **79.6** |
| | | TinyImageNet | 67.1 | 97.7 | **99.3** |
| | | LSUN | 67.0 | 97.9 | **99.6** |
| ResNet | CIFAR-10 | SVHN | 0.29 | **97.9** | 93.5 |
| | | TinyImageNet | 64.3 | 94.8 | **98.0** |
| | | LSUN | 67.9 | 98.6 | **99.3** |
| | CIFAR-100 | SVHN | 0.94 | 50.9 | **57.0** |
| | | TinyImageNet | 77.3 | 81.8 | **94.6** |
| | | LSUN | 81.7 | 80.0 | **97.0** |
| | SVHN | CIFAR-10 | 53.2 | **95.8** | 82.2 |
| | | TinyImageNet | 68.6 | **99.6** | 94.2 |
| | | LSUN | 68.9 | **99.6** | 92.4 |
| | Average | | 54.6 | 89.3 | **91.7** |
| | SD | | 31.4 | 12.4 | **10.7** |
| | Min | | 0.20 | 50.9 | **57.0** |

to the total deviation metric. Finally, all normalized deviations are summed together to produce a final abnormality score per sample.

Using the above formulation, we aim to translate these components into similar counterparts within our proposed framework. This allows us to evaluate the importance of each component, which is difficult to perform using the original formulation.

Namely, we split the spatial reduction into 2 separate functions. The first uses only the gram matrix (power-1) reduction denoted as $GP_1$, while the second uses either $GP_1$ or max pooling in conjunction with a modified version of the deviation function ($\delta^*$). Specifically, instead of measuring the maximum and minimum values over the training data, we choose to use less extreme quantiles (0.05 and 0.95). The new deviation function is defined as follows,

$$\delta^*(\text{quantile}_{0.05}, \text{quantile}_{0.95}, \text{value}) = \begin{cases} 0 & \text{quantile}_{0.05} \leq \text{value} \leq \text{quantile}_{0.95} \\ \frac{\text{quantile}_{0.05}-\text{value}}{|\text{quantile}_{0.05}|} & \text{value} < \text{quantile}_{0.05} \\ \frac{\text{value}-\text{quantile}_{0.95}}{|\text{quantile}_{0.95}|} & \text{value} > \text{quantile}_{0.95} \end{cases}.$$

(11)

The channel reduction remains unchanged, i.e. channels are reduced using a simple summation over the observed values.

For the layer reduction is replaced with the fisher statistic which is better suited for the task of exposing irregularities over all layers compared to simple summation. This also allows us to avoid using $E_{va}(\delta_l)$ normalization, by converting the observed layer deviations with their representing $p$-value. This leads to a uniform scale across all layers deviations.

We compare these variants on OOD detection tasks, where the main objective is to identify which elements are more dominant in their importance to the detection performance. From Table 7, we observe that $\delta^*$ is crucial for OOD detection. We attribute its importance to normalizing the channels according to their magnitudes before adding them to the layer deviation score. On the other hand, when replacing the Gram matrix with a simple max-pooling, the detector is marginally better on average. Based on our evaluation, we posit that the success of the GRAM method could potentially be based more on its deviation function than the specific use of Gram matrix countering the original intuition linked to style transfer. However, it is interesting to see $GP_1$ clearly outperforms max pooling in certain cases and vice versa.

## G.3 EXTENDED EVALUATION

We provide the results from Table 2 including the AUROC in Table 8 as well as AUROC curve in Fig. 8. We suggest that the AUROC metric is not well suited for OOD detection. In essence, it equally weighs TPR for all FPR, while lower T1E rates are predominantly relevant in practical applications. Furthermore, in Table 10 we report results for additional types of OOD intended to test the stability of the method (Peng et al., 2019; Zhou et al., 2017; Clanuwat et al., 2018; Xiao et al., 2017; LeCun et al., 1998). In these results, we used the entire test split of each of the additional OOD datasets.

We compare our method against GRAM as the best available observer method and MSP as a baseline. Since GRAM did not report their results on these OOD datasets, we provide our best effort to reproduce their results. We rely on the authors' open-sourced code with a slight modification. The main difference is that we do not rely on test data when constructing the test statistic for GRAM. This is done by splitting the training dataset into two splits. The first is a 90% split, used for extracting GRAM's min/max values per layer. The remaining 10% split is used for estimating the expectation over the deviation score (see Sastry and Oore (2019)). In addition, we replace the cross-validation with 5 random seeds and report the mean and (SD).

Table 10, shows GRAM and MaSF have comparable power. MaSF generally seems to have an advantage when more samples are available (SVHN in-dist), while GRAM seems to have a slight advantage when fewer samples are available (CIFAR-100 in-dist). It can be seen that both MaSF and GRAM are relatively weak compared to MSP in the near-distribution experiments (i.e., CIFAR-10-100). Conversely, MSP fails in "easy" scenarios, such as Normal random noise in TPR95 and AUROC. These results indicate that using multiple layers is important to obtain greater power in many scenarios; however, it might be harmful in near-distributions.

In Table 11 we report results for MSP, GRAM and MaSF using Wide-ResNet (Zagoruyko and Komodakis, 2016) models trained with Outlier Exposure (OE) (Hendrycks et al., 2018). OE is a training procedure that aims to reduce the Softmax over-confidence property on OOD samples. This is done by training the model parameters to maximize the entropy of the output for OOD samples. A large set of OOD data is used as a proxy for the unseen OOD test distribution. We use the models which were trained from "scratch" by the authors with Wu et al. (2017) as the exposed outlier dataset. MSP-OE results for CIFAR-10-100 scenarios are better than both GRAM and MaSF. Notably, MSP-OE significantly improves some of the catastrophic failures of MSP observed on "easy" OOD samples, such as the Normal random noise scenario. An expected improvement is also visible for datasets that are similar to the OE distribution, such as TinyImageNet. However, MSP-OE detection capacity is still limited compared to GRAM and MaSF for general OOD scenarios. Furthermore, we observe some of the weaknesses of MaSF. Specifically when CIFAR-100 is in-distribution. We associate this failure with the disproportion between the number of channels and the number of samples available.

Overall, these results expose some potential weaknesses of MaSF. However, we remind the reader that it is a relatively simple detector designed for efficient inference at a fraction of the compute budget of GRAM. Moreover, the "free" MSP alternative is appealing only for near distributions, while the trend of failures on seemingly "easy" OOD samples is not yet resolved. Thus, we conclude that as of date, there are no uniformly powerful detectors. Detecting unknown test distributions remains a challenging task. Our framework attempts to support this effort by allowing to design and combine multiple specialized detectors, using principled procedures borrowed from classic statistical hypothesis testing — while allowing the designer to maintain a meaningful error rate.

## G.4 ASSIGNING LAYER PRIORITIES

It is clear from the experimental results that certain combinations of in-out distribution datasets (e.g., CIFAR10 vs. CIFAR100 and vice versa) are more challenging than others. It is also apparent that simple methods that only use the final layer sometimes produce superior results in this scenario while failing on seemingly simple cases (e.g., Table 10, Normal random inputs). We suggest this failure is linked to the hierarchical processing in neural networks where the final layers target more complex features related to specific classes. In contrast, earlier layers target basic features common to the data seen during training and less class specific.

Table 8: Full Table 2, including AUROC

| Network | In-dist | Out-of-dist | TPR at 95% | | | | | AUROC | | | | |
|---|---|---|---|---|---|---|---|---|---|---|---|---|
| | | | MSP | Mahalanobis | Resflow | GRAM | MaSF | MSP | Mahalanobis | Resflow | GRAM | MaSF |
| DenseNet | CIFAR-10 | SVHN | 40.3 | 89.6 | 86.1 | 96.1 | **98.4** | 89.9 | 97.6 | 97.3 | 99.1 | **99.6** |
| | | TinyImageNet | 59.4 | 94.9 | 96.1 | **98.8** | 97.8 | 94.2 | 97.5 | 99.1 | **99.7** | 99.3 |
| | | LSUN | 66.9 | 97.2 | 98.1 | **99.5** | 99 | 95.5 | 98.3 | 99.5 | **99.9** | 99.4 |
| | CIFAR-100 | SVHN | 26.2 | 62.2 | 48.9 | **89.3** | 83.7 | 82.6 | 85.6 | 87.9 | **97.3** | 96.1 |
| | | TinyImageNet | 17.4 | 87.2 | 91.5 | **95.7** | 93.9 | 71.8 | 92.7 | 98.1 | **99.0** | 98.3 |
| | | LSUN | 16.6 | 91.4 | 95.8 | **97.2** | 97.2 | 71.0 | 95 | 98.9 | **99.3** | 99.1 |
| | SVHN | CIFAR-10 | 61.7 | **97.5** | 90.0 | 80.4 | 86.8 | 92.3 | 96.7 | **98** | 95.5 | 97.3 |
| | | TinyImageNet | 80.4 | **99.9** | 99.9 | 99.1 | 99.8 | 95.5 | 99.5 | **99.9** | 99.7 | 99.6 |
| | | LSUN | 80.2 | **100** | 100 | 99.5 | 99.9 | 95.5 | 99.8 | **99.9** | 99.8 | 99.6 |
| ResNet | CIFAR-10 | SVHN | 27.9 | 75.8 | 91.0 | 97.6 | **99** | 89.3 | 97.4 | 98.2 | 99.5 | **99.8** |
| | | TinyImageNet | 42.2 | 95.5 | 98.0 | **98.7** | 98.4 | 90.3 | 97.9 | 99.6 | **99.7** | 99.5 |
| | | LSUN | 41.3 | 98.1 | 99.1 | 99.6 | **99.7** | 90.1 | 99.2 | 99.8 | **99.9** | 99.8 |
| | CIFAR-100 | SVHN | 15.0 | 41.9 | 74.1 | 80.8 | **89.7** | 76.1 | 93.2 | 95.1 | 96.0 | **96.9** |
| | | TinyImageNet | 17.6 | 70.3 | 77.5 | 94.8 | **96.1** | 73.4 | 76.9 | 90.1 | **98.9** | 98.8 |
| | | LSUN | 15.0 | 56.6 | 70.4 | 96.6 | **98.2** | 70.9 | 66.2 | 87.2 | 99.2 | 99.2 |
| | SVHN | CIFAR-10 | 79.1 | 94.1 | 96.6 | 85.8 | **98** | 93.0 | 98.1 | 99 | 97.3 | **99.2** |
| | | TinyImageNet | 79.8 | 99.2 | **99.9** | 99.3 | 99.9 | 93.5 | 99.4 | **99.9** | 99.7 | 99.8 |
| | | LSUN | 75 | 99.9 | **100** | 99.6 | 100 | 91.5 | 99.9 | **100** | 99.8 | 99.8 |
| | Average | | 46.8 | 86.2 | 89.6 | 94.9 | **96.4** | 86.5 | 93.9 | 97.1 | 98.9 | **99.0** |
| | SD | | 26.0 | 16.9 | 13.8 | 6.4 | **4.8** | 9.4 | 9.0 | 4.2 | 1.4 | **1.1** |
| | Minimum | | 15.0 | 41.9 | 48.9 | 80.4 | **83.7** | 70.9 | 66.2 | 87.2 | 95.5 | **96.1** |

Naturally, when testing the global null, all $p$-values are combined with an even weight. This is since we do not know which hypothesis will be more significant without assuming any knowledge about the test distribution. We demonstrate the impact of assigning a larger weight on the final layer, reducing the contribution of the remaining layers uniformly. This can be achieved by replacing the standard Fisher test statistic s.t. $T_{\text{Fisher\_weighted}}(\boldsymbol{q}) = -2 \sum_{i=1}^{m} \alpha_i \ln(q_i)$ where $\alpha_i \geq 0, \quad \sum_{i=1}^{m} \alpha_i = 1$.

In Table 9 we observe that assigning a higher weight to the last layer can improve the detector's performance. Specifically, MaSF performance improves in the near-distribution experiments (CIFAR-100/10) while it deteriorates in the general case (i.e., vs. baseline from Table 2). Notably, when assigning a high weight to the final layer TPR 95% drops to 0 for in-dist CIFAR-100 and the ResNet, while the AUROC maintains a smoother decline. This indicates that the produced p-values resolution is not sufficient to separate between OOD and in-distribution samples.

This is due to the discrete percentile estimation and the inherent conservativeness of p-value retrieval, which is required to maintain the statistical guarantees of the method. Namely, the retrieved p-values are conservative in the sense that they will always be larger than the true p-value. For example, suppose we observe a realization of $4.01$ for a specific random variable, where the estimated $0.04$ quantiles is $4$ and the $0.05$ quantile is $5$, the assigned p-value (left-sided) would be $0.05$ even though the realization is much closer to $4$. Bounding the p-value in such a way ensures the validity of the method for any type of distribution. However, this reduces the power of the test. Even more so, when the number of observations is small since the percentiles are estimated sparsely.

In our experiments, the final layer has 100 channels (to match the number of classes), while the number of samples is per 500. Thus, the p-values resolution is insufficient to discriminate between in-dist and OOD as all samples receive the lowest possible percentile. As a result, at FPR of $5\%$, TPR is reduced to 0. AUROC behavior suggests that this gap closes for higher FPR rates. Furthermore, results suggest there seems to be a potential balance that benefits both near and general scenarios compared to vanilla MaSF. We leave the topic of weight assignment to future work.

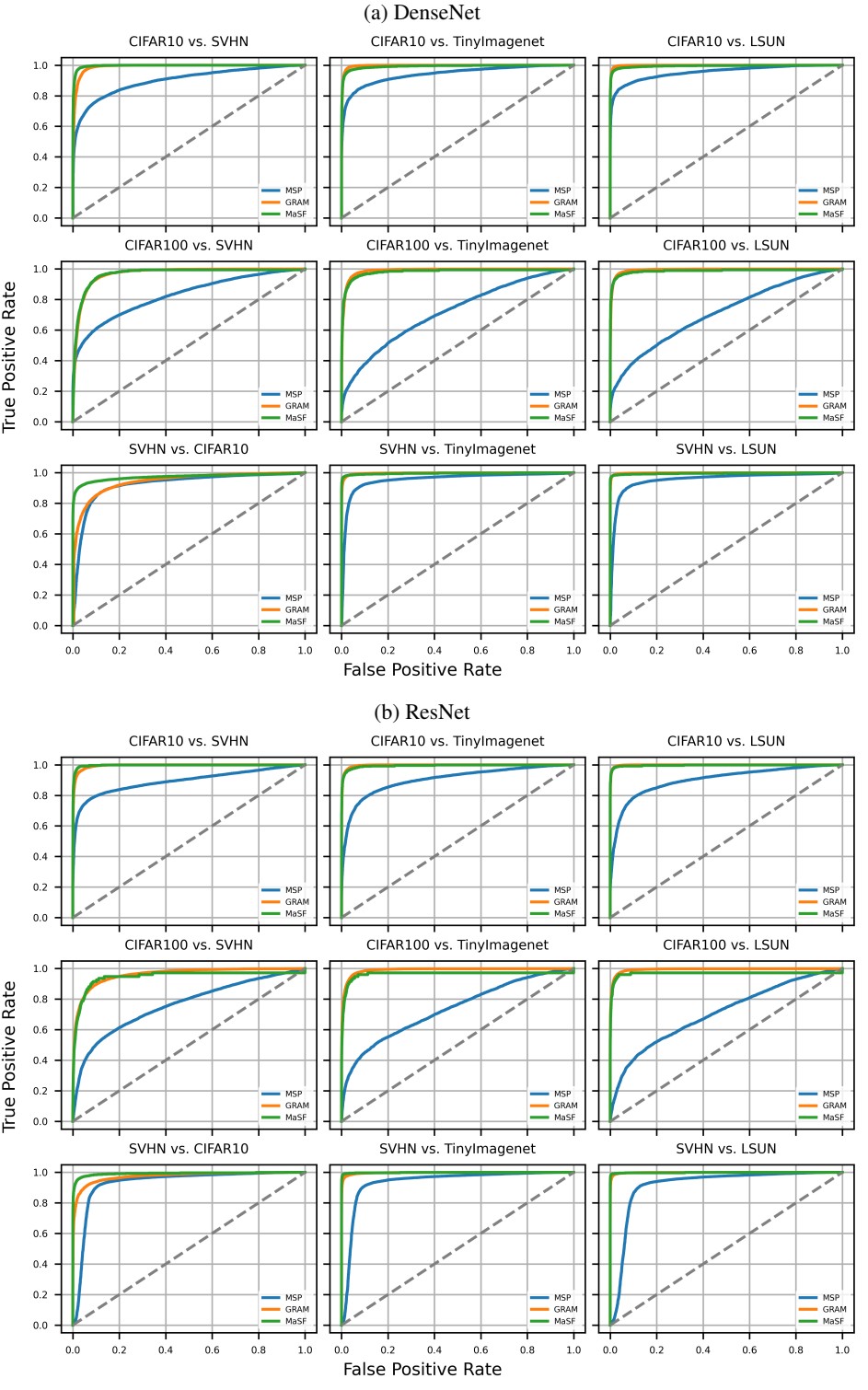

Figure 8: ROC plots for Table-2, MaSF typically yields better TPR for small FPR budget (<0.1).

Table 9: Comparison of weighted MaSF using varying weights on the final fully-connected layer. MSP as reference to the discriminative power of the networks' prediction based on this layer

| Network | In-dist | Out-of-dist | MSP | TPR at 95% Weighted MaSF | | | | | MSP | AUROC Weighted MaSF | | | | |
|---|---|---|---|---|---|---|---|---|---|---|---|---|---|---|
| | | | | 0.05 | 0.1 | 0.25 | 0.5 | 0.75 | | 0.05 | 0.1 | 0.25 | 0.5 | 0.75 |
| | | | | | Lee et al. (2018) benchmark. | | | | | | | | | |
| DenseNet3 | CIFAR-10 | SVHN | 40.3 | 98.7 | **99.1** | 98.5 | 92.4 | 80 | 89.9 | 99.7 | **99.8** | 99.7 | 98.1 | 94.9 |
| | | TinyImagenet | 59.4 | 98.5 | **98.9** | 96.5 | 88.7 | 83.7 | 94.2 | 99.4 | **99.5** | 99.3 | 97.3 | 95.3 |
| | | LSUN | 66.9 | 99.4 | **99.6** | 98.6 | 95.3 | 93.2 | 95.5 | 99.6 | 99.6 | **99.7** | 98.7 | 97.7 |
| | SVHN | CIFAR-10 | 61.8 | 89.4 | **91.4** | 88.9 | 68.2 | 40.2 | 92.3 | 97.5 | **97.7** | 97.5 | 95.6 | 93 |
| | | TinyImagenet | 80.4 | **99.8** | **99.8** | 99.5 | 92.5 | 57.5 | 95.5 | **99.6** | **99.6** | 99.5 | 98.4 | 95 |
| | | LSUN | 80.2 | **100** | **100** | 99.5 | 90.4 | 52.2 | 95.5 | **99.6** | **99.6** | 99.4 | 98 | 94.3 |
| | CIFAR-100 | SVHN | 26.2 | 84.7 | 85.9 | **87** | 78.5 | 46.6 | 82.6 | 96.3 | 96.5 | **96.6** | 95.3 | 90 |
| | | TinyImagenet | 17.4 | **93.6** | **93.6** | 89.5 | 72.5 | 52.2 | 71.8 | 98.4 | 98.3 | 97.7 | 94.4 | 89.2 |
| | | LSUN | 16.6 | 97.1 | **97.2** | 95.6 | 81.5 | 59.6 | 71 | **99.1** | **99.1** | 98.8 | 96.6 | 92.3 |
| ResNet34 | CIFAR-100 | SVHN | 15 | **90.4** | 89.5 | 87.8 | 0 | 0 | 76.2 | 97.1 | **97.2** | 96 | 90 | 83.7 |
| | | TinyImagenet | 17.6 | **96.2** | 95.6 | 85.5 | 0 | 0 | 73.4 | **98.8** | 98.5 | 95.9 | 87.1 | 79 |
| | | LSUN | 15 | **98** | 97.5 | 84 | 0 | 0 | 70.9 | **99.2** | 98.9 | 95.8 | 86.3 | 78.5 |
| | SVHN | CIFAR-10 | 79.1 | **98.1** | 97.8 | 96 | 91.6 | 88.9 | 93 | **99.2** | **99.2** | 98.9 | 98 | 96.8 |
| | | TinyImagenet | 79.9 | **99.9** | **99.9** | 99.3 | 95.7 | 90.5 | 93.5 | **99.8** | **99.8** | 99.6 | 98.9 | 97.4 |
| | | LSUN | 75 | **100** | 99.9 | 99 | 93.2 | 86.6 | 91.5 | **99.8** | **99.8** | 99.6 | 98.5 | 96.8 |
| | CIFAR-10 | SVHN | 27.9 | **99** | 98.9 | 97.3 | 85 | 54.5 | 89.3 | **99.8** | 99.7 | 99.3 | 96.9 | 91.8 |
| | | TinyImagenet | 42.2 | **98.4** | 97.9 | 94.6 | 82.8 | 74 | 90.3 | **99.5** | **99.5** | 98.9 | 96.6 | 94.1 |
| | | LSUN | 41.3 | **99.6** | 99.5 | 98.1 | 92 | 86.8 | 90.1 | **99.8** | **99.8** | 99.4 | 97.9 | 96.3 |
| | | Avg | 46.8 | 96.7 | **96.8** | 94.2 | 72.2 | 58.1 | 86.5 | **99** | **99** | 98.4 | 95.7 | 92 |
| | | SD | 25.3 | 4.2 | **4** | 5.3 | 33.1 | 30.7 | 9.1 | **1** | **1** | 1.4 | 3.8 | 5.8 |
| | | Min | 15 | 84.7 | **85.9** | 84 | 0 | 0 | 70.9 | 96.3 | **96.5** | 95.8 | 86.3 | 78.5 |
| | | | | | Near-distribution CIFAR-10-100 | | | | | | | | | |
| DenseNet3 | CIFAR-10 | CIFAR-100 | 40.6 | 23.7 | 30.4 | 40.9 | **46.4** | 46.2 | **89.3** | 76.2 | 80.9 | 85.4 | 85.5 | 85.2 |
| | CIFAR-100 | CIFAR-10 | **18.4** | 4.8 | 5.5 | 6 | 6.5 | 4.8 | **75.8** | 54.5 | 55.8 | 57.2 | 57.1 | 56.6 |
| ResNet34 | CIFAR-100 | CIFAR-10 | **17.6** | 6.1 | 7 | 4.2 | 0 | 0 | **76.6** | 66.2 | 69.4 | 73.6 | 74.3 | 74 |
| | CIFAR-10 | CIFAR-100 | 33.7 | 34.7 | 37.8 | **41.8** | 40.6 | 39.5 | **86.4** | 83 | 84.4 | 86.1 | 86.2 | 85.8 |
| | | Avg | **27.6** | 17.3 | 20.2 | 23.2 | 23.1 | 22.6 | **82** | 70 | 72.6 | 75.6 | 75.8 | 75.4 |
| | | SD | **9.9** | 12.5 | 14.2 | 18.2 | 20.6 | 20.4 | **5.9** | 10.7 | 11.2 | 11.7 | 11.8 | 11.8 |
| | | Min | **17.6** | 4.8 | 5.5 | 4.2 | 0 | 0 | **75.8** | 54.5 | 55.8 | 57.2 | 57.1 | 56.6 |

Table 10: Extended evaluation of MSP, GRAM and MaSF for DenseNet and Resnet models. R, C indicate resize and crop respectively.

| Network | In-dist | Out-of-dist | TPR at 95% | | | AUROC | | |
| --- | --- | --- | --- | --- | --- | --- | --- | --- |
| | | | MSP | GRAM | MaSF | MSP | GRAM | MaSF |
| DenseNet3 | CIFAR-10 | CIFAR-100 | **40.6** | 26.3 (0.4) | 18.9 | **89.3** | 72.2 (0.1) | 69.7 |
| | | Infograph | 44.6 | 89.8 (0.4) | **89.9** | 91 | 97.8 (0.1) | **98** |
| | | Quickdraw | 11.4 | **100 (0)** | **100** | 84 | **100 (0)** | **100** |
| | | Sketch | 41.5 | 93.6 (0.3) | **94.5** | 90.5 | 98.6 (0) | **98.7** |
| | | Fashion-MNIST | 80.5 | 99.9 (0) | **100** | 97.3 | **99.9 (0)** | 99.7 |
| | | LSUN (C) | 52.1 | **87.7 (0.4)** | 83 | 93 | **97.3 (0.1)** | 96.3 |
| | | LSUN (R) | 66.9 | **99.4 (0)** | 99 | 95.5 | **99.9 (0)** | 99.4 |
| | | Textures | 49.1 | **89.9 (0.3)** | 88.1 | 91.5 | **97.9 (0.1)** | 97.6 |
| | | TinyImageNet (C) | 57 | **96.3 (0.2)** | 93.3 | 93.8 | **99.2 (0)** | 98.4 |
| | | TinyImageNet (R) | 59.4 | **98.6 (0.1)** | 97.8 | 94.2 | **99.7 (0)** | 99.3 |
| | | K-MNIST | 79.1 | **100 (0)** | **100** | 97 | **100 (0)** | 99.9 |
| | | MNIST | 75.3 | **100 (0)** | **100** | 96.8 | **100 (0)** | 99.9 |
| | | Places-365 | 52.9 | **72.2 (0.6)** | 64.1 | 93.1 | **93.4 (0.1)** | 92.2 |
| | | $N(0,1)$ | 74.4 | **100 (0)** | **100** | 96.4 | **100 (0)** | **100** |
| | | SVHN | 40.3 | 95.8 (0.1) | **98.4** | 89.9 | 99.1 (0) | **99.6** |
| | CIFAR-100 | CIFAR-10 | **18.4** | 7.3 (0.2) | 4.5 | **75.8** | 59.4 (0.2) | 52.9 |
| | | Infograph | 37.9 | **74.1 (0.4)** | 67.8 | 84.3 | **93.8 (0.1)** | 90.7 |
| | | Quickdraw | 90.3 | **100 (0)** | **100** | 97.8 | **100 (0)** | **100** |
| | | Sketch | 39.8 | **78.6 (0.5)** | 73.6 | 83.6 | **94.6 (0.1)** | 92.6 |
| | | Fashion-MNIST | 64.9 | 99.2 (0.1) | **99.7** | 92.7 | **99.6 (0)** | 99.5 |
| | | LSUN (C) | 28.4 | **60.5 (0.4)** | 47.8 | 80.1 | **89.5 (0.1)** | 82.7 |
| | | LSUN (R) | 16.6 | **97.4 (0.1)** | 97.2 | 71 | **99.4 (0)** | 99.1 |
| | | Textures | 23.4 | **64 (0.5)** | 53.6 | 76.8 | **90.3 (0.1)** | 85.6 |
| | | TinyImageNet (C) | 24.4 | **87.6 (0.3)** | 80.8 | 76.2 | **97.2 (0)** | 95.1 |
| | | TinyImageNet (R) | 17.4 | **95.3 (0.1)** | 93.9 | 71.8 | **98.9 (0)** | 98.3 |
| | | K-MNIST | 33.9 | **100 (0)** | **100** | 83.1 | **99.9 (0)** | 99.8 |
| | | MNIST | 33.2 | **100 (0)** | **100** | 84 | 99.8 (0) | **99.9** |
| | | Places-365 | 28.3 | **35.1 (0.3)** | 21.7 | 80.7 | **80.1 (0.1)** | 68.8 |
| | | $N(0,1)$ | 0 | **100 (0)** | **100** | 36.7 | **100 (0)** | **100** |
| | | SVHN | 26.2 | **87.7 (0.4)** | 83.7 | 82.6 | **96.8 (0.1)** | 96.1 |
| | SVHN | CIFAR-10 | 61.8 | 68.6 (1.6) | **86.8** | 92.3 | 94.1 (0.3) | **97.3** |
| | | CIFAR-100 | 61.4 | 75 (1.3) | **88** | 91.9 | 95.5 (0.2) | **97.4** |
| | | Infograph | 55.2 | 97 (0.2) | **98.6** | 89.1 | 99.3 (0) | **99.4** |
| | | Quickdraw | 19 | **100 (0)** | **100** | 76.1 | **100 (0)** | **100** |
| | | Sketch | 49.5 | 96.5 (0.1) | **98.2** | 86.4 | 99.2 (0) | **99.4** |
| | | Fashion-MNIST | 99.5 | **100 (0)** | **100** | 99.2 | **100 (0)** | 99.9 |
| | | LSUN (C) | 67.1 | 91.7 (0.4) | **95.4** | 92.8 | 98.2 (0.1) | **98.9** |
| | | LSUN (R) | 80.2 | 99.6 (0.1) | **99.9** | 95.5 | **99.9 (0)** | 99.6 |
| | | Textures | 64.4 | 90.2 (0.3) | **91.6** | 91.9 | 97.9 (0) | **98.3** |
| | | TinyImageNet (C) | 76.9 | 97.1 (0.3) | **99** | 95.1 | 99.3 (0.1) | **99.4** |
| | | TinyImageNet (R) | 80.4 | 99.1 (0.2) | **99.8** | 95.5 | **99.7 (0)** | 99.6 |
| | | K-MNIST | 100 | **100 (0)** | **100** | 99.1 | **100 (0)** | **100** |
| | | MNIST | 100 | **100 (0)** | **100** | 99 | **100 (0)** | **100** |
| | | Places-365 | 68.2 | 84.5 (0.9) | **92** | 93.1 | 96.9 (0.1) | **98.2** |
| | | $N(0,1)$ | 98.7 | **100 (0)** | **100** | 97.2 | **100 (0)** | **100** |
| ResNet34 | CIFAR-10 | CIFAR-100 | 33.7 | **38.3 (0.6)** | 32.4 | **86.4** | 82.2 (0.3) | 82.1 |
| | | Infograph | 53.3 | 92 (0.3) | **94.8** | 92.8 | 98.4 (0) | **98.9** |
| | | Quickdraw | 71.2 | **100 (0)** | **100** | 92.8 | **100 (0)** | 99.9 |
| | | Sketch | 55.3 | 94 (0.3) | **96.4** | 92.6 | 98.9 (0.1) | **99.2** |
| | | Fashion-MNIST | 45.3 | 98.6 (0.3) | **99.8** | 89.8 | 99.5 (0.1) | **99.7** |
| | | LSUN (C) | 46 | 90.1 (0.3) | **91.8** | 91.7 | 98 (0) | **98.3** |
| | | LSUN (R) | 41.3 | 99.4 (0.1) | **99.7** | 90.1 | **99.9 (0)** | 99.8 |
| | | Textures | 37 | 89.7 (0.5) | **91.8** | 88.7 | 98.1 (0.1) | **98.5** |
| | | TinyImageNet (C) | 44 | 96 (0.3) | **96.4** | 91 | **99.2 (0)** | 99.1 |
| | | TinyImageNet (R) | 42.2 | 98.3 (0.1) | **98.4** | 90.3 | **99.7 (0)** | 99.5 |
| | | K-MNIST | 41.1 | **100 (0)** | **100** | 90.5 | **99.9 (0)** | **99.9** |
| | | MNIST | 32.9 | **100 (0)** | **100** | 87.5 | **99.9 (0.1)** | **99.9** |
| | | Places-365 | 40.1 | 76.4 (0.6) | **78.7** | 90.3 | 95.4 (0.1) | **96** |
| | | $N(0,1)$ | 83.1 | **100 (0)** | **100** | 96.9 | **100 (0)** | **100** |
| | | SVHN | 27.9 | 97.1 (0.3) | **99** | 89.3 | 99.4 (0) | **99.8** |
| | CIFAR-100 | CIFAR-10 | **17.6** | 9.6 (0.3) | 5.7 | **76.6** | 69.3 (0.2) | 64 |
| | | Infograph | 21 | 76.3 (0.1) | **76.5** | 76.7 | **95.3 (0)** | 94.7 |
| | | Quickdraw | 11.8 | **100 (0)** | **100** | 68.1 | **99.9 (0)** | 99.7 |
| | | Sketch | 21.3 | **80.2 (0.3)** | 79.6 | 76.6 | **95.8 (0)** | 95.2 |
| | | Fashion-MNIST | 36.9 | 98 (0.2) | **99.6** | 86.4 | 99.3 (0) | **99.4** |
| | | LSUN (C) | 16 | **63.6 (0.4)** | 59.9 | 74.1 | **92.2 (0.1)** | 90.1 |
| | | LSUN (R) | 15 | 97.5 (0) | **98.2** | 70.9 | **99.4 (0)** | 99.2 |
| | | Textures | 17.8 | 66.5 (0.5) | **70** | 75.3 | **92.4 (0.1)** | 92 |
| | | TinyImageNet (C) | 21.9 | **88.9 (0.1)** | 88.8 | 77.2 | **97.7 (0)** | 97.4 |
| | | TinyImageNet (R) | 17.6 | 95.5 (0.1) | **96.1** | 73.4 | **99 (0)** | 98.8 |
| | | K-MNIST | 26.4 | 99.9 (0) | **100** | 82.1 | **99.7 (0)** | **99.7** |
| | | MNIST | 17.9 | **100 (0)** | **100** | 74.6 | 99.6 (0) | **99.7** |
| | | Places-365 | 19.6 | **41.7 (0.5)** | 32.2 | 77.8 | **86.6 (0.2)** | 81.1 |
| | | $N(0,1)$ | 3.4 | **100 (0)** | **100** | 77.1 | **100 (0)** | 99.7 |
| | | SVHN | 15 | 80.8 (1.2) | **89.7** | 76.1 | 96.1 (0.1) | **96.9** |

| In-dist | Out-of-dist | | | | | | |
|---|---|---|---|---|---|---|---|
| | CIFAR-10 | 79.1 | 86.8 (0.4) | **98** | 93 | 97.4 (0.1) | **99.2** |
| | CIFAR-100 | 78 | 87.4 (0.3) | **97.7** | 92.7 | 97.5 (0.1) | **99.1** |
| | Infograph | 81 | 96.7 (0.1) | **99.3** | 94.1 | 99.2 (0) | **99.7** |
| | Quickdraw | 95.5 | **100 (0)** | 100 | 97.2 | **100 (0)** | 99.9 |
| | Sketch | 80.2 | 97.1 (0.1) | **99.2** | 93.9 | 99.4 (0) | **99.7** |
| | Fashion-MNIST | 98.3 | **100 (0)** | 100 | 99.1 | **100 (0)** | 99.9 |
| | LSUN (C) | 78 | 93.6 (0.2) | **98.7** | 93 | 98.6 (0) | **99.5** |
| SVHN | LSUN (R) | 75 | 99.6 (0) | 100 | 91.5 | **99.8 (0)** | 99.8 |
| | Textures | 81.8 | 95.5 (0.2) | **98.4** | 94.4 | 99 (0) | **99.5** |
| | TinyImageNet (C) | 81.2 | 98.2 (0.1) | **99.8** | 94.2 | 99.4 (0) | **99.8** |
| | TinyImageNet (R) | 79.8 | 99.3 (0.1) | **99.9** | 93.5 | 99.7 (0) | **99.8** |
| | K-MNIST | 98.2 | **100 (0)** | 100 | 99.1 | **100 (0)** | 99.9 |
| | MNIST | 96.4 | **100 (0)** | 100 | 98.5 | **100 (0)** | 99.9 |
| | Places-365 | 76.9 | 89.2 (0.3) | **97.6** | 92.6 | 97.7 (0.1) | **99.2** |
| | $N(0,1)$ | 85.8 | **100 (0)** | 100 | 96.2 | **100 (0)** | 99.9 |
| Avg | | 51.4 | 88.3 | **88.5** | 87.7 | **96.8** | 96.3 |
| SD | | 27.5 | **19.6** | 21.6 | 9.8 | **6.7** | 8.1 |
| Min | | 0.0 | **7.3** | 4.5 | 36.7 | **59.4** | 52.9 |

Table 11: Extended evaluation with Outlier Exposure (Hendrycks and Gimpel, 2016) using Wide-ResNets. R,C indicate resize and crop respectively.

| | | TPR at 95% | | | AUROC | | |
|---|---|---|---|---|---|---|---|
| In-dist | Out-of-dist | MSP | GRAM | MaSF | MSP | GRAM | MaSF |
| | CIFAR-100 | **73.8** | 49 (1.3) | 25.3 | **94.8** | 83.3 (0.4) | 76.9 |
| | Infograph | **99.1** | 96.5 (0.3) | 93.9 | **99.7** | 99.2 (0) | 98.7 |
| | Quickdraw | 94.5 | **100 (0)** | 100.0 | 98.9 | **100 (0)** | 100.0 |
| | Sketch | **98.8** | 97 (0.2) | 96.0 | **99.6** | 99.3 (0) | 99.1 |
| | Fashion-MNIST | 91.4 | 99.5 (0.1) | **100.0** | 98.5 | **99.7 (0.1)** | 99.7 |
| | LSUN (C) | **98.0** | 92.8 (0.4) | 85.7 | **99.4** | 98.4 (0.1) | 97.1 |
| | LSUN (R) | 98.5 | **99.6 (0)** | 99.4 | 99.4 | **99.9 (0)** | 99.7 |
| CIFAR-10 | Textures | 97.1 | 94.4 (0.3) | 91.2 | **99.2** | 98.8 (0) | 98.3 |
| | TinyImageNet (C) | 95.2 | **96.8 (0.1)** | 95.5 | 98.7 | **99.2 (0)** | 99.0 |
| | TinyImageNet (R) | 93.8 | **98.7 (0.1)** | 98.5 | 98.5 | **99.7 (0)** | 99.6 |
| | K-MNIST | 97.1 | **100 (0)** | 100.0 | 99.1 | **99.9 (0)** | **99.9** |
| | MNIST | 92.8 | **100 (0)** | 100.0 | 98.5 | **99.9 (0)** | **99.9** |
| | Places365 | **97.3** | 85.2 (0.9) | 67.5 | **99.3** | 96.7 (0.2) | 93.7 |
| | $N(0,1)$ | **100.0** | **100 (0)** | 100.0 | 98.8 | **100 (0)** | 100.0 |
| | SVHN | 98.0 | 98.1 (0.1) | **98.2** | 99.5 | **99.5 (0)** | **99.5** |
| | CIFAR-10 | **20.1** | 8.5 (0.2) | 3.5 | **79.5** | 61.8 (0.2) | 51.4 |
| | Infograph | **85.4** | 84.6 (0.3) | 71.7 | **97.2** | 96.6 (0.1) | 93.1 |
| | Quickdraw | 98.4 | **100 (0)** | 100.0 | 99.2 | **100 (0)** | 99.9 |
| | Sketch | 84.0 | **88.6 (0.4)** | 76.3 | 96.9 | **97.5 (0.1)** | 94.2 |
| | Fashion-MNIST | 81.9 | **98.1 (0.2)** | 96.6 | 96.6 | **99.5 (0.1)** | 99.0 |
| | LSUN (C) | 72.9 | **73.3 (0.4)** | 53.3 | **94.6** | 94 (0.1) | 86.4 |
| | LSUN R | 54.3 | **98.5 (0.1)** | 98.1 | 89.3 | **99.6 (0)** | 99.5 |
| CIFAR-100 | Textures | 67.3 | **77.5 (0.2)** | 64.6 | 93.3 | **94.9 (0)** | 90.3 |
| | TinyImageNet (C) | 42.6 | **91 (0.2)** | 86.6 | 85.0 | **98 (0.1)** | 96.7 |
| | TinyImageNet (R) | 34.6 | **96.5 (0.1)** | 95.7 | 81.9 | 99.2 (0) | **99.0** |
| | K-MNIST | 82.6 | 99.9 (0) | **100.0** | 96.7 | **99.9 (0)** | **99.9** |
| | MNIST | 58.7 | **100 (0)** | 100.0 | 92.4 | 99.8 (0) | **99.9** |
| | Places-365 | **70.7** | 51.9 (0.4) | 26.3 | **94.5** | 88 (0.1) | 74.5 |
| | $N(0,1)$ | 71.2 | **100 (0)** | 100.0 | 96.0 | **100 (0)** | 99.9 |
| | SVHN | 65.5 | **84.9 (0.3)** | 81.3 | 94.1 | **96.7 (0.1)** | 95.3 |
| | CIFAR-10 | **100.0** | 99.9 (0) | **100.0** | **100.0** | 99.8 (0.1) | 99.9 |
| | CIFAR-100 | **100.0** | 99.8 (0) | **100.0** | **100.0** | 99.8 (0.1) | 99.9 |
| | Infograph | **100.0** | **100 (0)** | **100.0** | **100.0** | 99.9 (0) | **100.0** |
| | Quickdraw | **100.0** | **100 (0)** | **100.0** | **100.0** | 99.8 (0) | 99.9 |
| | Sketch | **100.0** | 99.9 (0) | **100.0** | **100.0** | 99.8 (0.1) | 99.9 |
| | FashionMNIST | **100.0** | **100 (0)** | **100.0** | **100.0** | 99.9 (0) | **100.0** |
| | LSUN (C) | **100.0** | **100 (0)** | **100.0** | **100.0** | 99.9 (0) | **100.0** |
| SVHN | LSUN (R) | **100.0** | **100 (0)** | **100.0** | **100.0** | **100 (0)** | **100.0** |
| | Textures | **100.0** | 99.7 (0) | **99.9** | **100.0** | 99.9 (0) | 99.9 |
| | TinyImageNet (C) | **100.0** | **100 (0)** | **100.0** | **100.0** | **100 (0)** | **100.0** |
| | TinyImageNet (R) | **100.0** | **100 (0)** | **100.0** | **100.0** | **100 (0)** | **100.0** |

| | | | | | | |
|---|---|---|---|---|---|---|
| K-MNIST | **100.0** | **100 (0)** | **100.0** | **100.0** | **100 (0)** | **100.0** |
| MNIST | **100.0** | **100 (0)** | **100.0** | 99.9 | **100 (0)** | **100.0** |
| Places-365 | **100.0** | **100 (0)** | **100.0** | **100.0** | 99.9 (0.1) | 99.9 |
| $N(0, 1)$ | **100.0** | **100 (0)** | **100.0** | **100.0** | **100 (0)** | **100.0** |
| Avg | 87.0 | **92.4** | 89.0 | 97.1 | **97.7** | 96.4 |
| SD | 19.4 | **17** | 21.9 | 4.7 | 6.3 | **8.7** |
| Min | **20.1** | 8.5 | 3.5 | **79.5** | 61.8 | 51.4 |

