# OpenReview forum: "A Statistical Framework for Efficient Out of Distribution Detection in Deep Neural Networks"
_ICLR.cc/2022/Conference — ICLR 2022 Poster_

### Official Review · Reviewer_XHJz · 2021-10-28

**Correctness:** 3
**Technical Novelty And Significance:** 3
**Empirical Novelty And Significance:** 3
**Recommendation:** 8
**Confidence:** 3

**Main Review:**

Their framework is interesting and solves a relevant problem. It also seems to work well as, based on results, it outperforms baseline methods in several cases being also much more computationally light.

However, a problem of the manuscript is that it quite extensively uses appendices as extra space (the whole paper is 28 pages long). Their framework is not completely described or cannot be understood if appendices are left out. I agree that improving this is a very hard task, but e.g. moving Algorithm listing (page 21) to the main body would already provide better understanding of the framework.

Specific comments:

- There are few instances in which authors are used statistical terminology that are unclear for wider audience. For instance, ".. maintaining the Type 1 Error .." gives impression that the aim is to keep T1E error level in certain level.

- I might have missed this point, but how was the threshold for p value for final decision (Algorithm 1 output) chosen?

- page 3, "... the p-value is given by $q(t_{obs})=P(...)$ ...", this would be less confusing if it would be mentioned that p-values are denoted by q

- Eq (1): I is not explained, I guess it's indicator function

- Section 3.2: the procedure of obtaining p-values was not unclear based in this description (as I do not have too background in hypothesis testing). Perhaps the content of footnote 3 and/or Algorithm description from appendix could be moved to this section?

- Could "Related work" be earlier? The baseline methods are now described here, but already used in the previous section.

- Second paragraph of discussion mentions that OoD detector cannot be constructed without any assumptions regarding the test distribution. I am bit puzzled that how this is reflecting to the proposed framework. Is it that there is assumption that training data for the framework comes from the same distribution as the testing and results are more or less arbitrary if not? But, in the experiments, the OOD detector was trained using the same training data as CNN (i.e. in-distribution data?) without seeing any out-of-distribution samples?

- page 13, "F convlutional neural network"

- page 18, figure 3: what red and blue colors represent? Font is also quite small

- page 21, Algorithm 1, the second last line: $T^L$ is taking $t_1^c$s as input, but based on (6), the input of $T^L$ is p-values. Algorithm is missing t->q calculation in this point?


**Summary Of The Paper:**

The paper proposes a new framework to detect out of distribution samples for deep neural networks without retraining or modifying the model. The method is based on statistical hypothesis testing which is applied to activations of all or multiple layers.


**Summary Of The Review:**

As the framework is novel and results shows good performance both in accuracy and computational complexity, I feel that the study deserves to be published. But the length might be issue.

I chose "marginally above the acceptance threshold" as my recommendation due to the length issue. If there would not be any limits, my recommendation would be "accept, good paper".

---

> ### Author Response · Authors · 2021-11-18
> **Response to XHJz**
>
> We thank the reviewer for acknowledging the contribution of our paper and its relevancy.
>
> Regarding the length of the paper:
> A significant effort was made to distill all the required knowledge into the main paper. The appendix entries generally aim to complement claims and arguments made in the main paper and are not needed to understand the main contributions.
> Based on this feedback we moved the algorithm (in addition to the Simes and Fisher test statistic functions) into the main paper and will be happy to receive any further ideas on how to improve it.
>
> Use of statistical terminology:
> We provide a brief introduction to NHST and explain the relevant nomenclature. However, we must assume the readers have some basic knowledge to fit the paper within the page limit. Regarding the T1E comment, it is correct. Specifically, maintaining the T1E is an important topic of this paper since it is not possible to guarantee the test power for arbitrary OOD detection (i.e., only T1E tolerance can be controlled).
>
> Rejection policy:
> In our case, a threshold of 5% is used (implying the T1E probability is less than 5%).
> In general, the threshold is selected by the user according to the tolerance for T1E. More specifically, since the p-value is stochastically larger than uniform (proposition 1,2), any cut-off used will correspond to the probability of making a T1E.
>
> P-value notation:
> A clarification was added. We consistently use T as a test statistic function, t as the value of the test statistic, and q as the p-value.
>
> Indicator notation:
> You are correct. we have clarified it in the revision to eliminate doubt.
>
> Clarity of section 3.2:
> We have added a short paragraph with a high-level overview of the calibration procedure to complement the technical description and moved the algorithm up from the appendix. Note that there is also a qualitative summary in section 3.3. In essence, each hierarchy of p-values requires estimating the appropriate test statistic eCDFs using the appropriate dataset splits per class by applying the eCDF operator defined in eq. 1.
>
> Related work location:
> We received mixed sentiment from our early readers. The section appears later to avoid exhausting the readers before reaching the core of the paper. We added a clear reference to related work in the revision, if this seems crucial to the reviewer we are willing to move the related work section before section 4.
>
> Assumptions on OOD at test time:
> In our framework, we only use in-distribution samples and assume nothing on the outlier test distribution. The second paragraph refers to a theoretical result (Birnbaum, 1954) that a “uniformly most powerful test” cannot be constructed without prior knowledge of the actual test distribution (i.e., samples from both in-dist and out-dist that are used to test the detector). That is, any single detector cannot outperform all other detectors in every possible test distribution. the citation appears earlier when discussing the empirical results, we added a citation here as well.
>
> Misc:
> Page 13 typo was corrected. Figure 3 describes the magnitude of the correlation between layers: Red is positive, and blue is negative. A description was added to the figure caption.
>
> Algorithm 1 comment:
> In general, you are correct. However, Algorithm 1 describes MaSF where $T^\mathrm{ch}$ is a multiple comparison method (Simes test). Thus, $t^c$ is already a p-value that can be used for $T^L$ (see Appendix E). We have addressed this in the revision by adding a relevant transition within the algorithm to avoid confusion.
>
> We hope our reply addressed the reviewer's concerns, if so we would appreciate it if the reviewer would consider increasing the rating despite the length of the appendix.

---

### Official Review · Reviewer_Ni17 · 2021-11-01

**Correctness:** 3
**Technical Novelty And Significance:** 2
**Empirical Novelty And Significance:** 2
**Recommendation:** 5
**Confidence:** 5

**Main Review:**

Strengths:

(1) Different from previous work, the paper considers the empirical distribution of each layer and channel in CNN and proposes to use global null tests with Simes and Fisher statistics to aggregate the p-values.

(2) The proposed method is more computationally efficient than compared baselines while preserving comparable performances.

(3) The effect of the proposed method on different layers and channels is discussed in detail.


Weaknesses:

(1) Proposition 1 and 2 are obtained by the underlying cumulative distribution without considering the gap between the estimated cumulative distribution and the real one. Equation (8) in the proof of proposition 1 is true only when q extracted is from the real distribution. It seems like it is considered as the real distribution in the following analysis, while the experiments consist of the estimated cumulative functions and corresponding p-values.

(2) The proposed method uses p-values from every layer and channel in CNN, but intuition and arguments of Fisher and Simes tests as layer and channel reduction methods are not fully presented. As the core of the proposed algorithm,  it would be better to include the background of Fisher/Simes tests and why they are effective as layer and channel reduction respectively.  The current draft only includes a brief introduction with no discussion in the appendix.

(3) It is great to see the computational cost is much lower than baselines such as Gram and Mahalanobis, but some baselines do not require retrain or expensive computation (e.g. Energy score (Liu et al., 2020)) are missing. It would be interesting to see the comparative performance, both computation-wise and detection accuracy-wise.

Suggestions for improvement

(1) The notation of t and T can be pre-defined before introducing equation (4) (rather than presented in the appendix) to facilitate better understanding.

(2) Move the introduction of Fisher and Simes to the main paper.

(3) Given the large OOD uncertainty space, it's desirable to also include more evaluation datasets as used in literature.

(4) Add more recent baselines, especially the ones that are already computationally efficient.

**Summary Of The Paper:**

This paper constructs OOD detection as a hypothesis testing problem and extracts p-values from each layer and channel in CNN from the empirical cumulative distribution. Max, Simes, and Fisher test statistics are adopted as spatial, channel, and layer reduction methods (MaSF). Empirical results show that MaSF achieves comparable or better results than GRAM (Sastry and Oore, 2019), ResFlow (Zisselman and Tamar, 2020), and Mahalanobis (Lee et al., 2018) while significantly reducing the computational cost.

**Summary Of The Review:**

Hypothesis testing using the empirical cumulative distribution is not novel enough as a "new framework": existing works already include such spirit of it by using FPR95 of the in-distribution samples. The theoretical part is also not strong enough without considering the gap between the true and estimated cumulative distribution.

The proposed method uses p-values from every layer and channel in CNN, but intuition and arguments of Fisher and Simes tests as layer and channel reduction methods are not fully presented. Lastly, the experimental evaluations can be further strengthened (by adding more recent baselines).

---

> ### Author Response · Authors · 2021-11-18
> **Response to Ni17**
>
> The novelty of the general statistical framework:
> Please note that applying a threshold over an unnormalized score to fix T1E is only equivalent to statistical guarantees on T1E when the output score does not depend on the predicted class (which is not the case in some SOTA detectors such as GRAM). In addition, methods that output (valid) p-values also allow to maintain statistically meaningful error rates in downstream applications (e.g., T1E when combining multiple detectors or FDR in batch testing scenarios).
> In our work, we explicitly formulate the relevant hypotheses and provide the required correction in the class-conditional case (i.e., proposition 2) to produce valid class-conditional p-values (shown both theoretically and empirically).
> Moreover, our framework extends prior work based on conformal predictors by showing how to aggregate multiple tests together in a hierarchical fashion. This contribution is empirically shown to improve the detector power by 50% on average compared to single-layer variants (section 4.2).
>
> The gap between the true and eCDF:
> We apologize if this was not clear. The theoretical guarantee is with respect to the true distribution (not only the empirical one). Specifically, we mentioned in appendix B.1 (just above the assumptions) that the proof builds on Vovk 2005 (proposition 4.1). This guarantees that the conformal p-value obtained from the eCDF is stochastically larger than the real p-value. It relies on the fact that $t^c(X_{test}; \chi_{train})$ is exchangeable with $\{ t^c(X; \chi_{train}^c): X \in \chi_{val}^c\}$. To remove any confusion we added the relevant step in our proofs.
>
> Intuition and arguments of Fisher and Simes tests:
> We provide a self-contained introduction to Simes and Fisher tests in appendix B.3.  Moreover, please note that we also provide significant insight to using Simes and Fisher tests when applied to NHST in DNNs, see sections 3.3 (P5:L1-8, global testing with many hypotheses) and 4.2 (under subsection “choosing reductions” P5:L(-2)-P6:L4) when discussing the construction of MaSF.
> In a nutshell, we suggest Simes and Fisher are appropriate choices for channel and layer reductions as OOD evidence tends to propagate throughout the network and is likely to either be sparse over channels or that the signal is strongly correlated between channels within the same layer.
> We added an explicit description of the functionality of these tests for a clearer description in relation to their properties in section 2 of the revised paper.
>
> Comparison to Liu et. al. 2020 energy-score (ES):
> ES works well as a “mutator” method (i.e., fine-tuning on an OOD dataset before testing). This procedure requires additional hyperparameters (HP) and a potentially costly HP search, while the final model accuracy is not guaranteed. In addition, the detection performance can also degrade on outlier distributions that are dissimilar to the proxy dataset (as we discuss in our paper’s intro).
> Notably, this setting is fundamentally different from our own, as we assume no prior knowledge on the test distribution and do not alter the model weights. Moreover, when the ES is applied without fine-tuning (observer settings), its results severely degrade (see tables 1 and 2 in the aforementioned paper).
> Specifically, ODIN and Mahalanobis detector (Lee et. al. 2018), appear to be strong competitors. For instance, see WideResNet-CIFAR100 in table 2. The mean FPR (T1E) of Mahalanobis is 54% vs. 73.6% for ES (while OOD detection power is fixed at 95%). A direct comparison cannot be drawn due to differences between the sets of experiments in each paper. However, we show significantly better results than Lee et. al. 2018. For instance, in our Table-1 Mahalanobis mTPR is 86.1% vs. 96.4% for MaSF (while the T1E is fixed at 5%). We added a reference for the paper under related work.
>
> Comparison to other efficient baselines:
> To the best of our knowledge, the reported baselines represent SOTA with respect to observer methods. Please let us know if there are other recent relevant examples.
>
> More evaluation datasets:
> Please note that we provide an extensive evaluation in appendix G.3 (see table 10).
>
> We hope our answers satisfy the reviewer’s concerns regarding the novelty of the proposed framework, the correctness of theoretical results, and the extent of method evaluation. We would appreciate additional feedback if not.

---

> > ### Author Response · Authors · 2021-11-18
> > **Additional remark on the computational cost and test power compared to Liu 2020 (energy-score ES):**
> >
> > We suggest that the power of the ES as OOD detector is expected to be similar to the Fisher test statistic over the output logits’ p-values (i.e., using fisher test as channel reduction in a single layer detector), at a similar test time computational cost.
> >
> > However, the energy score does not normalize the contribution of individual logits with respect to their distribution before aggregation. This can potentially degrade results as large values mask smaller ones. In addition, the sum of exponential terms can lead to high variance and numerical instability as the number of classes increases. We present tightly related discussions with respect to Fisher’s test statistic, these can be found in sections 3.3, 4.2, and appendix B3.

---

> > ### Author Response · Authors · 2021-11-30
> > **A gentle reminder**
> >
> > Please note our response and updated revision. If there are any remaining concerns please let us know, and if not, we kindly ask the reviewer to consider raising the score

---

### Official Review · Reviewer_a8KZ · 2021-11-03

**Correctness:** 3
**Technical Novelty And Significance:** 2
**Empirical Novelty And Significance:** 2
**Recommendation:** 3
**Confidence:** 5

**Main Review:**

Strengths:

1. Considering OOD detection as a hypothesis test is an interesting formulation.

2. Proposed approach considers an 'Observer'-style approach that does not require updating the underlying model. This is more efficient and easily accessible.

3. Proposed framework achieves high efficiency while detecting OOD samples and can be deployed in real-time setups.

Weaknesses:

1. Formulating OOD detecting as a hypothesis test is previously explored in the following works. What are the novel contributions compared to these works. Authors are encouraged to add these in the reference section.

a. Haroush et al., "Statistical Testing for Efficient Out of Distribution Detection in Deep Neural Networks"

b. Cai & Koutsoukos, "Real-time Out-of-distribution Detection in Learning-Enabled Cyber-Physical Systems"

c. Bates et al., "Testing for Outliers with Conformal p-values"

2. Experiments are limited in terms of comparisons with the recent state-of-the-art approaches and evaluating on benchmark datasets. Here are a few recent approaches on OOD detection and authors are encouraged to compare with these approaches.

a. Hendrycks et al., "Deep anomaly detection with outlier exposure"

b. Liu et al., "Energy-based out-of-distribution detection"

c. Lee eta al., "Training confidence-calibrated classifiers for detecting out-of-distribution samples"


**Summary Of The Paper:**

This paper poses detecting out of distribution (OOD) samples as a hypothesis testing problem. Specifically, the authors propose a variant of the inductive conformal prediction framework where a 'p'-value is computed as a measure for OOD-ness. Finally, the OOD samples are detected by thresholding the 'p'-values.

**Summary Of The Review:**

The approach lacks novelty and experiments are limited to justify the efficacy of the proposed approach.

---

> ### Author Response · Authors · 2021-11-18
> **Response to a8KZ**
>
> We thank the reviewer for the feedback.
> Please note that we cannot cite the first paper mentioned, as it is the previous draft of this paper (i.e., the same one that is being reviewed here).
>
> Regarding the other papers mentioned:
> Cai & Koutsoukos’s work belongs to a parallel domain, where OOD detection is not conditioned on a given predictor or objective other than OOD detection. In our case, the prediction is explicitly conditioned on a specific pre-trained network. These two domains are somewhat similar, yet incompatible for comparison (note that they also do not reference any other related work in our domain).
>
> Bates et al. work is orthogonal to our paper, moreover, there is no reason to believe its application for OOD detection in DNNs will perform better than MSP. Specifically, it describes a methodology that extends the theory behind conformal inference (Vovk 2005, i.e. how to compute conformal p-values). This approach produces stronger statistical guarantees on errors at the expense of test power. Moreover, Bates et. al. complements our proposed framework as a plug-in replacement to the standard conformal p-values, in the case T1E is more important than test power. Yet, this is not always the case with OOD detection. Finally, we deal with the class-conditional case while Bates and Vovk focus on one-class predictors. We added a short paragraph in the revision to explain this connection.
>
> Regarding MaSF evaluation:
> Generally, the suggested papers do not belong to the relevant class of detectors (i.e., mutators), nor perform better than the methods we considered (which to the best of our knowledge, represent SOTA observer methods). In addition, the OOD detection evaluation presented in our paper is far more extensive in terms of architectures and datasets (see appendix G.3).
>
> More specifically, these “mutators” methods rely on training the classifier on OOD data (or a proxy), which can degrade performance on alternative outlier data distribution. We reference Hendrycks et. al. as part of the related work despite it being a “mutator” method due to its impact on later papers and present related results in appendix G.
> Finally, Liu et al. can be applied as an observer method as well, however, its results without “fine-tuning” on OOD data are clearly inferior to other baselines used in our work such as Lee et al. 2018.  For instance, see WideResNet-CIFAR100 in table 2 of the aforementioned paper. The mean FPR (T1E) of Mahalanobis is 54% vs. 73.6% for energy-score (while OOD detection power is fixed at 95%). A direct comparison cannot be drawn due to differences between the sets of experiments in each paper. However, we show significantly better results than Lee et. al. 2018. For instance, in our Table-1 Mahalanobis mTPR is 86.1% vs. 96.4% for MaSF (while the T1E is fixed at 5%).
> Given the above, we kindly ask the reviewer to reconsider her/his position regarding the novelty of the paper and the given score.

---

> > ### Author Response · Authors · 2021-11-30
> > **A gentle reminder**
> >
> > Please note our response and the updated revision. If there are any remaining concerns please let us know, and if not, we kindly ask the reviewer to consider raising the score

---

### Official Review · Reviewer_zW6V · 2021-11-07

**Correctness:** 3
**Technical Novelty And Significance:** 4
**Empirical Novelty And Significance:** 4
**Recommendation:** 8
**Confidence:** 3

**Details Of Ethics Concerns:**

None.

**Main Review:**

### Strengths

S1) The paper frames the out-of-distribution (OOD) detection problem in the framework of hypothesis testing, which more directly links the problem to existing statistical tools and methods.

S2) The paper proposes a new, computationally efficient method (MaSF) for performing hypothesis testing which uses multiple layers from a deep neural network. The authors report many empirical experiments which supports their method.

S3) The paper is generally well-written.


### Weaknesses

**W1) Lack of comparison / discussion about "calibration" literature**

I am admittedly not very familiar with the OOD detection literature. However, the motivation for OOD detection, as described in the paper's introduction, is that deep neural networks output prediction scores that tend to be overconfident. This sounds like a problem of "calibration," which the paper does not mention. For example, a canonical paper on calibration of deep neural networks is "[On Calibration of Modern Neural Networks](http://proceedings.mlr.press/v70/guo17a.html)" by Guo et al. 2017. I would appreciate a discussion of how OOD detection is related too and/or different from calibration techniques, given that they are motivated by the same underlying problem of overconfident neural network predictions.

**W2) Applicability to regression settings?**

The method described seems to be specifically designed for classification problems. Is there a similar formulation for regression problems? If not, I think it should be emphasized (maybe even in the paper title, e.g., "A Statistical Framework for Efficient Out of Distribution Detection in Deep Neural Network Classifiers") that the method is specific for classification.

**W3) Needs more background / justification for Fisher and Simes tests**

Perhaps this is due to my unfamiliarity with the Fisher and Simes tests, but I found the paper lacking in justification for why these were the two tests that authors use. What about other spatial reduction and channel reduction functions, such as simple mean/max reductions? What if you used a neural network for the reduction functions?

One way that the authors could help with this issue is to describe the Fisher and Simes tests in the body of the main text, instead of leaving it for the appendix.

**W4) Implications for designing neural networks?**

Are there any implications of the OOD detection method for designing better neural networks that are able to generalize better? For example, does this method present any natural mechanism for better calibrating neural network layers?

**W5) Other clarity issues**

- In Section 1.1 (Contributions), the definition of a Type I Error seems wrong, or at least poorly worded. The paper says that a Type I Error is _detecting OOD as in-distribution_, which to me is confusing. A better wording could be, for example, _incorrectly predicting "OOD" for an actual in-distribution sample_.
- In Section 3.1 (Preliminaries), the definition for $N_{val}$ seems to be a typo. I think it should say $N_{train}$.
- Why is the global null hypothesis denoted with an intersection $\cap$ symbol? Is each $H_{0,i}$ a set? And if so, wouldn't it be a union instead of an intersection? Presumably this is notation that is standard in the literature, so pardon my ignorance. In any case, I think it would be helpful to clarify the notation in the text.

**Summary Of The Paper:**

The authors formalize the out-of-distribution (OOD) detection problem in the framework of statistical hypothesis testing. The proposed method, called Max-Simes-Fisher (MaSF), is designed for deep neural networks and combines sample features extracted from multiple layers in the network. The method returns a $p$-value for every test sample at a specified significance level (_i.e._, probability of incorrectly predicting "OOD" for an actual in-distribution sample). Compared with existing methods for OOD detection, the proposed method achieves similar or better True Positive Rates (TPR, _i.e._, correct OOD prediction rate) with significantly lower computation cost.

**Summary Of The Review:**

I preface this summary with an admission that I am not the most familiar with the OOD detection literature and global null hypothesis testing. However, I believe that formulation of OOD detection as a hypothesis testing problem is well thought out and beneficial for the ML community by rooting the problem in a concrete statistical framework. Furthermore, the proposed method seems sound (I skimmed through the math, albeit not thoroughly) and has significant speed-ups compared to existing methods, such that the method is far more likely to be deployed in real-world applications and edge devices. For these reasons, I believe this to be a valuable contribution to the ML community.

---

> ### Author Response · Authors · 2021-11-18
> **Response to zW6V**
>
> We thank the reviewer for acknowledging the contributions in our paper and for the insightful feedback.
>
> Discussion on calibration vs. OOD detection:
> We added a comment within the introduction. OOD detection is connected to uncertainty/calibration in the sense that we would expect well-calibrated models to assign low confidence scores for OOD samples (in fact we would want these scores to be 0.).
> Yet, calibration literature aims to assign confidence scores that correspond with the probability of model error while operating on data drawn from similar distribution to the one seen during training (i.e., a correct prediction exists). In contrast, OOD detection deals with rejecting OOD samples altogether (no correct answer exists). Therefore datasets, benchmarks, and metrics designed for one space are generally irrelevant to the other.
>
> Application to regression tasks:
> Our framework can easily apply in the context of regression, as the regressed prediction can always be translated into scalar p-values (e.g., the intermediate layers’ features can be seen as predictions for a regression problem over the model’s latent spaces). In the context of classification, we measure class-dependent statistics to obtain more powerful test statistics. A similar approach could be used by assigning meta-classes to groups of inputs of a regression task (e.g., via clustering) or by using a one-class approach. We aim to explore this subject further in future work. We added the appropriate comment in the revised manuscript.
>
> Simes and Fisher missing justification:
> First, note that the general framework allows for arbitrary choice of reductions. Next, we provide a self-contained introduction to Simes and Fisher tests in appendix B.3.
> Moreover, please note that we provide significant insight to using Simes and Fisher tests when applied to NHST in DNNs, see sections 3.3 (P5:L1-8, global testing with many hypotheses) and 4.2 (under subsection “choosing reductions” P5:L(-2)-P6:L4) when discussing the construction of MaSF.
> In a nutshell, we suggest Simes and Fisher are appropriate choices for channel and layer reductions as OOD evidence tends to propagate throughout the network and is likely to either be sparse over channels or that the signal is strongly correlated between channels within the same layer.
> We added an explicit description of the functionality of these tests for a clearer description in relation to their properties in section 2 of the revised paper.
>
> Using other reductions:
> Learning dedicated reductions could be possible. However, such an approach is more expensive and will probably be challenging to train end to end. In contrast, Mean/Max/Fisher/Simes are efficient parameter-free that work well. We also experiment with other reductions such as the Mahalanobis distance and outer-product in appendix G.
> Applications for designing and training neural networks:
> p-values in general can be used as an auxiliary to the model prediction. We find that small p-values correlate with model errors, while correct predictions are uniformly distributed under the null hypothesis (i.e., training and test are drawn from the same distribution). Our specific method can be more effective as it incorporates information from all layers (i.e., not only the last). However, this direction requires further research.
> We believe there are many potential applications for NHST in DNNs. Our method could be used to “mine” difficult inputs and create curriculum-based/active-learning training regimes or even filter outliers from large-scale datasets. Another less trivial application could be to improve robustness by actively suppressing outlier channels or as a guide for pruning methods by observing the affinity between pairs of certain channels and classes (e.g., avoid pruning highly discriminative channels). Along similar lines, one can also use our framework to determine what layers should be retrained in transfer learning scenarios according to the distribution of the new data p-values with respect to the original training model.
> These are all ideas that we intend to explore in future work.
>
> Intersection notation:
> The intersection of $H_{0,i}$ implies that all of the null hypotheses are correct, the alternative is that at least one null hypothesis is false. This notation is fairly standard, we define it earlier in section 2 under the global null test subsection.
>
> Misc:
> T1E wording: T1E wording was changed to be more precise.
> Nval typo: You are correct, this was fixed in the revision.

---

### Decision · Program_Chairs · 2022-01-20

**Decision:**

Accept (Poster)

**Comment:**

The paper considers the empirical distribution of layer/channel in CNN ,and proposes to use global null tests with Simes and Fisher statistics to aggregate the p-values. This method is competitive while computationally efficient. The underlying theoretical insights are discussed in detail.

The paper received mixed ratings, and the discussions weren't active. So, AC carefully read the paper and inspected all reviews. Reviewer a8KZ comments were factually inaccurate in listing references, and lack substantial feedback on the actual content of the paper. Hence, the review was down-weighted.

The other negative reviewer Ni17, as an OoD expert, unfortunately did not offer more feedback to author rebuttals. From what AC comprehends, the authors should have clarified their The theoretical guarantee and compared properly with Liu et. al. 2020 energy-score (ES).

Considering the above, AC feels that the study deserves to be published.